# Talin-activated vinculin interacts with branched actin networks to initiate bundles

Rajaa Boujemaa-Paterski[1,2]*, Bruno Martins[1], Matthias Eibauer[1], Charlie T Beales[1], Benjamin Geiger[3], Ohad Medalia[1]*

[1]Department of Biochemistry, University of Zurich, Zurich, Switzerland; [2]Université Grenoble Alpes, Grenoble, France; [3]Department of Immunology, Weizmann Institute of Science, Rehovot, Israel

**Abstract** Vinculin plays a fundamental role in integrin-mediated cell adhesion. Activated by talin, it interacts with diverse adhesome components, enabling mechanical coupling between the actin cytoskeleton and the extracellular matrix. Here we studied the interactions of activated full-length vinculin with actin and the way it regulates the organization and dynamics of the Arp2/3 complex-mediated branched actin network. Through a combination of surface patterning and light microscopy experiments we show that vinculin can bundle dendritic actin networks through rapid binding and filament crosslinking. We show that vinculin promotes stable but flexible actin bundles having a mixed-polarity organization, as confirmed by cryo-electron tomography. Adhesion-like synthetic design of vinculin activation by surface-bound talin revealed that clustered vinculin can initiate and immobilize bundles from mobile Arp2/3-branched networks. Our results provide a molecular basis for coordinate actin bundle formation at nascent adhesions.

## Introduction

Integrin-based cell adhesions mediate the interactions of the actin cytoskeleton with the extracellular matrix (ECM), enabling cell migration, proliferation and differentiation. Vinculin, a key adapter protein regulating cell adhesion signaling, contributes to tissue homeostasis, cell morphogenesis, immune processes, and wound healing (*Carisey and Ballestrem, 2011*). Composed of multiple 'adhesome' scaffolding and signaling components, integrin adhesions, for example focal adhesions (FAs), are associated with the termini of contractile actin and myosin-rich stress fibers (*Geiger, 1979*; *Horton et al., 2016*; *Zaidel-Bar et al., 2007a*). These assemblies develop in response to external or internal (mostly cytoskeletal) traction forces (*Balaban et al., 2001*; *Geiger et al., 2009*; *Livne et al., 2014*; *Vicente-Manzanares et al., 2009*). The mechanism whereby integrin activation leads to FA formation is rather complex, and involves a cascade of conformational transitions of key mechanoresponsive proteins such as talin, vinculin, and FA kinase (*Burridge and Connell, 1983*; *Carisey et al., 2013*; *del Rio et al., 2009*; *Grashoff et al., 2010*; *Kong et al., 2009*; *Legate et al., 2011*; *Roberts and Critchley, 2009*; *Webb et al., 2004*; *Yang et al., 2016*; *Zamir and Geiger, 2001*). During nascent adhesion formation, talin changes its conformation to enable integrin binding (*Jiang et al., 2003*; *Wegener et al., 2007*). Next, force-sensing proteins such as vinculin are recruited and activated through its tail (*Burridge and Mangeat, 1984*; *del Rio et al., 2009*; *Hemmings et al., 1996*; *Hu et al., 2016*), leading to a series of molecular interactions involving adhesome scaffolding and signaling components (*Horton et al., 2016*; *Zaidel-Bar et al., 2007a*). Furthermore, the association of vinculin with both talin and the actin cytoskeleton generates tension that, in turn, induces conformational changes in vinculin, further activating it (*Grashoff et al., 2010*). Thus, vinculin is thought to act as a molecular clutch that couples the actin network to nascent

*For correspondence:
r.boujemaa@bioc.uzh.ch (RB-P);
omedalia@bioc.uzh.ch (OM)

**Competing interests:** The authors declare that no competing interests exist.

adhesions and FAs (*Bachir et al., 2014*; *Case and Waterman, 2015*; *Choi et al., 2008*; *Thievessen et al., 2013*).

Particularly intriguing is the mode of interaction of vinculin with the actin cytoskeleton. In motile cells, vinculin is believed to be involved in the coupling of the retrograde flow of the lamellipodial branched actin network to nascent integrin adhesions, which controls their initiation, maturation and disassembly dynamics (*Case et al., 2015*; *Choi et al., 2008*; *Giannone et al., 2007*; *Oakes et al., 2012*; *Thievessen et al., 2013*; *Zaidel-Bar et al., 2007a*). Vinculin was shown to be a mechanosensitive partner involved in FA maturation at the lamellipodium-lamellum border, where it responds to actomyosin-induced tension by increasing growth and robustness of FAs and their associated stress fibers (*Balaban et al., 2001*; *Chrzanowska-Wodnicka and Burridge, 1996*; *Humphries et al., 2007*). Vinculin is a component not only of nascent adhesions and FAs, but also of adherens-type cell-cell junctions, where it also acts as a mechanosensing component (*Seddiki et al., 2018*; *Yonemura et al., 2010*).

Vinculin is a flexible multidomain protein comprising a seven-helical head domain, a proline-rich linker, and a five-helical bundle tail domain (*Bakolitsa et al., 2004*; *Borgon et al., 2004*; *Molony and Burridge, 1985*; *Winkler et al., 1996*). In its inactive state, it exists in an auto-inhibitory conformation in which the head interacts with the tail domain of the protein, thus masking many of its functional binding sites (*Borgon et al., 2004*; *Chorev et al., 2018*). Interaction of vinculin with activated talin, coupled with mechanical perturbation, results in an open vinculin conformation, which enables its binding to various adhesome and cytoskeletal proteins (*Bois et al., 2005*; *Brindle et al., 1996*; *Carisey and Ballestrem, 2011*; *Case et al., 2015*; *DeMali et al., 2002*; *Hüttelmaier et al., 1997*; *Jockusch and Isenberg, 1981*; *Johnson and Craig, 1994*; *Kim et al., 2016*; *Thompson et al., 2017*; *Zamir and Geiger, 2001*). While a large body of evidence confirms the pivotal role of vinculin as a mechanosensor and a transducer of actin-generated tension to integrin adhesions throughout their functional states, little is known of how vinculin interacts with and organizes the architecture of the actin cytoskeleton at these sites.

Here we designed an experimental system to assess interactions of dynamic actin networks with full-length vinculin, in solution and upon vinculin interaction with immobilized talin fragment, resembling an adhesion site. We utilized in vitro reconstitutions of purified proteins to determine the molecular mechanisms underlying vinculin-mediated bundle initiation from branched actin networks. Next, we analyzed the effects of soluble activated vinculin on both dense actin organizations and sparse filament networks, to resolve the effect of actin crowding on vinculin crosslinking. Using reconstituted patterned actin organizations, we showed that activated full-length vinculin slows down lamellipodial branched networks growth and binds cellular-like actin organizations with comparable affinities. Through single filament total internal reflection fluorescence (TIRF) microscopy, we provide evidence that vinculin remodels Arp2/3 complex-mediated dendritic actin networks. These interactions are dominated by rapid binding of vinculin and fast crosslinking of actin filaments (F-actin) into stable and flexible bundles. Spatial restriction on vinculin activation by surface-bound talin, indicated its ability to interact with mobile Arp2/3 complex-mediated branched actin networks, modify their organization and to initiate a complex bundle formation. Therefore, our results highlight the ability of vinculin to connect filaments from various actin organizational states and suggest that talin-associated vinculin engages lamellipodial branched networks, from which it can initiate the formation of a stable network of actin bundles at the integrin nascent adhesion sites.

## Results

### Talin-activated vinculin alters the organization and dynamics of lamellipodium-like Arp2/3 complex-mediated branched actin networks

The lamellipodial actin network is a dynamic and dense mesh made of short and branched filaments (*Svitkina and Borisy, 1999*) capable of opposing the front-edge membrane tension and inducing protrusions (*Pollard and Borisy, 2003*). The Arp2/3 complex is a canonical actin nucleator of the leading edge (*Mullins et al., 1998*; *Svitkina and Borisy, 1999*), that together with the capping protein αβ (CP, a physiological marker of lamellipodia) (*Iwasa and Mullins, 2007*; *Pollard and Borisy, 2003*), organize actin into stiff and propulsive dendritic networks (*Achard et al., 2010*; *Akin and Mullins, 2008*; *Loisel et al., 1999*). The Arp2/3 complex is constitutively inactive. It is recruited to

and activated at the plasma membrane by the WAVE signaling complex, where it assembles branched filaments that generate force and push the membrane forwards (*Machesky et al., 1999*; *Svitkina and Borisy, 1999*). The lamellipodial actin treadmills centripetally and was proposed to be coupled via vinculin to early stages of nascent adhesions (*Case and Waterman, 2015*; *Choi et al., 2008*; *Thievessen et al., 2013*; *Zaidel-Bar et al., 2003*). Yet, how vinculin-F-actin interactions modulate branched network dynamics remains unclear.

First, we expressed the full-length version of human vinculin in insect cells, since post-translational modifications might be relevant for its function (*Golji et al., 2012*). The activation of the full-length vinculin by the constitutively active vinculin-binding site 1 (VBS1, residues 482–636) of talin 1 (*Cohen et al., 2005*; *Papagrigoriou et al., 2004*) allowed its binding to F-actin. Next, we investigated the effect of vinculin-F-actin interactions with branched networks at a density and stiffness that resembles the cellular environment. Using two-dimensional patterns printed on a protein-repellent glass coverslip and functionalized with a nucleation promoting factor (NPF), a C-terminal fragment of the Wiskott-Aldrich syndrome protein (WASP-pWA), which activates Arp2/3 complex-mediated nucleation of dendritic actin networks. Due to its fast and stable interactions with the actin barbed end (*Schafer et al., 1996*), CP prevents the expansion of Arp2/3-mediated branches away from the nucleation patterns. Capped actin branches host de novo nucleation by Arp2/3 complex, thereby ensuring the formation of a rigid branched meshwork, the growth of which results in a backward motion of the entire network (*Figure 1A*, *Figure 1—video 1*; *Boujemaa-Paterski et al., 2017*). Thus, the addition of VBS1-activated vinculin to lamellipodium-like structures significantly reduced their growth (*Figure 1B*, *Figure 1—video 1*), in line with in vivo studies showing that adhesion-bound vinculin engages and halts lamellipodial retrograde flow (*Choi et al., 2008*; *Thievessen et al., 2013*).

Steady state analyses showed that activated vinculin distributed homogenously throughout the entire network (*Figure 1C*) with an apparent dissociation constant of 0.2 mM (*Figure 1D*), independent of the network density (*Figure 1E*). Surprisingly, increasing the concentration of vinculin resulted in networks with higher actin density (*Figure 1F*). This may reflect (i) a mesoscale effect that results from a reorganization of the dendritic network by vinculin, or (ii) a microscale, biochemical effect of vinculin either on the dendritic nucleation activity of Arp2/3 complex, as vinculin was shown to bind Arp2/3 complex in vivo (*Chorev et al., 2014*; *Chorev et al., 2018*; *DeMali et al., 2002*), or on actin polymerization dynamics (*Jannie et al., 2015*; *Le Clainche et al., 2010*; *Wen et al., 2009*).

## The influence of actin organization on vinculin-mediated bundling

Vinculin was shown to participate in bridging cell adhesions with lamellipodial and lamellar actin, of which architecture, dimensionality and density may vary greatly (*Geiger et al., 2009*; *Thievessen et al., 2013*; *Xu et al., 2012*). We therefore explored how local actin organization affects vinculin recruitment and its ability to bundle the filamentous networks. Patterns of dotted rings were produced for studying three main types of cellular-like actin organization; namely, branched actin networks (*Svitkina and Borisy, 1999*), filopodial-like uniform-polarity filament bundles (*Svitkina and Borisy, 1999*; *Vignjevic et al., 2006*), and mixed-polarity bundles resembling the contractile segments of stress fibers (*Hotulainen and Lappalainen, 2006*; *Xu et al., 2012*; *Figure 2*, *Figure 2—video 1*). We used a similar setting as described above, but omitted CP. This led branched filaments to be nucleated on functionalized patterns and grow unrestricted with their fast-growing ends directed outwards (*Figure 2A*).

VBS1-activated vinculin formed stable bundles composed of uniform- and mixed-polarity filaments, and bound branched F-actin networks (*Figure 2B*, *Figure 2—video 2*). We quantified the fluorescence intensity ratio (vinculin:actin) of the three different networks (*Figure 2—figure supplement 1*) assembled on >500 dotted patterned as well as on continuous patterned rings (*Figure 2C, D*). Here, high and low density actin networks, represented by high and low fluorescence intensity networks respectively, were exposed to identical vinculin concentrations and were analyzed. Control experiments showed that addition of VBS1 or vinculin alone has no effect on the actin organization (*Figure 2—figure supplement 2*). These experiments demonstrated that while vinculin binding to actin is independent of the actin organization (i.e. vinculin affinities are similar in branched, uniform, or mixed-polarity networks) it is sensitive to actin density. Namely, sparser actin networks showed a higher occupancy than the denser ones (*Figure 2D*) while dense lamellipodium-like networks that contain CP exhibit ~eightfold higher vinculin-actin dissociation constant (*Figure 1D*).

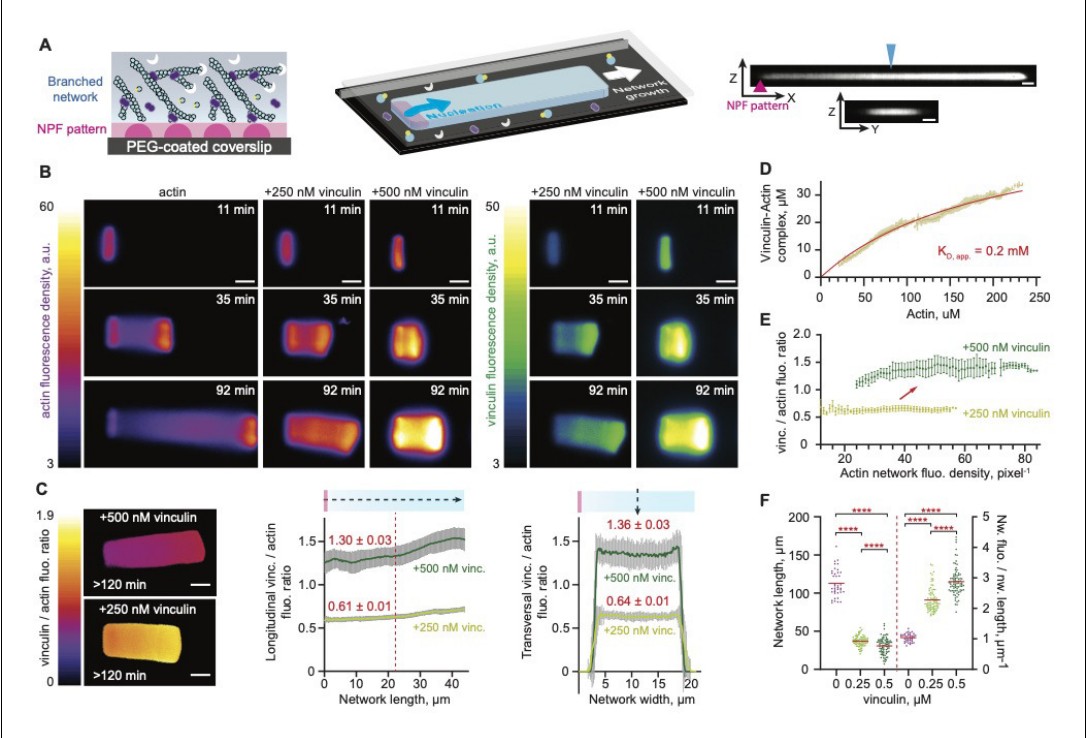

**Figure 1.** Vinculin alters the dynamics and the organization of lamellipodium-like branched actin networks. (**A**) A schematic illustration of dendritic network growth from a WASP pWA-micropattern on a two-dimensional glass coverslip (left) in the presence of profilin–actin (in blue-yellow), CP (in white), and Arp2/3 (in purple). The three-dimensionally constrained growth of lamellipodium-like branched actin network is forced by the geometry (middle scheme). Median, orthogonal ZY and ZX views from confocal imaging of a representative network is shown on the right. The blue arrowhead indicates the position of the orthogonal ZX section. The purple arrowhead indicates the position of the Nucleation promoting factor (NPF) pattern. Scale bars, 4 µm. (**B**) Wide-field epifluorescence images of the growth of different actin networks on 3 × 15 µm² GST-WASP pWA-coated patterned bar. Actin polymerized in a 10 µm high chamber in the presence of 4 µM G-actin Alexa-568 labeled, 8 µM Profilin, 120 nM Arp2/3 complex, 16 nM CP and talin VBS1-vinculin as indicated. (**C**) Analyses of longitudinal and transversal vinculin to actin fluorescence ratio along and across 25 networks, assembled as in B. Fluorescence intensities were measured along the line-scans (doted arrows) shown in the schematics. Plateau means are indicated. (**D**) Steady state saturation curve was fitted assuming mass-action binding at equilibrium. (**E**) Fluorescence quantification of the networks showed a stable vinculin to actin fluorescence ratio over the actin density. (**C, D, E**) Error bars represent the standard deviation from 25 networks per condition. (**F**) Dependence of networks length and actin fluorescence density on vinculin concentration. Statistical comparisons using Holm-Šídák test and a one-way analysis of variance (ANOVA) showed significant variations from 39, 83, 89 values for 0, 0.25, 0.5 µM vinculin, respectively, with p values < 0.0001. (**B, C**) Scale bars, 10 µm.

The online version of this article includes the following video and source data for figure 1:

**Source data 1.** Vinculin alters the dynamics and the organization of lamellipodium-like branched actin networks.

**Figure 1—video 1.** Vinculin slows down the dynamics of lamellipodium-like dendritic actin networks.

https://elifesciences.org/articles/53990#fig1video1

Since actin architectures polymerized by adhesion-associated actin nucleators can be attached to their nucleation sites (*Balaban et al., 2001*; *Choi et al., 2008*; *Chorev et al., 2014*; *DeMali et al., 2002*; *Lavelin et al., 2013*; *Legerstee et al., 2019*; *Oakes et al., 2012*), we tested how surface anchorage and filament density in the intermeshed, branched networks (on the patterns) affect the bundling. We utilized continuously patterned rings and polymerize an actin network of reduced branched density. In the absence of vinculin, isolated branched actin networks polymerized and intermeshed, resulting in smooth coverage of the patterns (*Figure 2—figure supplement 3A,B,E*; *Figure 2—video 1*). In the presence of VBS1-activated vinculin, dense networks polymerized on the rings were decorated by vinculin, and prominent bundles formed between the densely actin subnetworks (*Figure 2—figure supplement 3C,D,E*; *Figure 2—video 3*). These findings showed that vinculin not only binds branched and intermeshed actin filaments but reorganizes them into stable bundles.

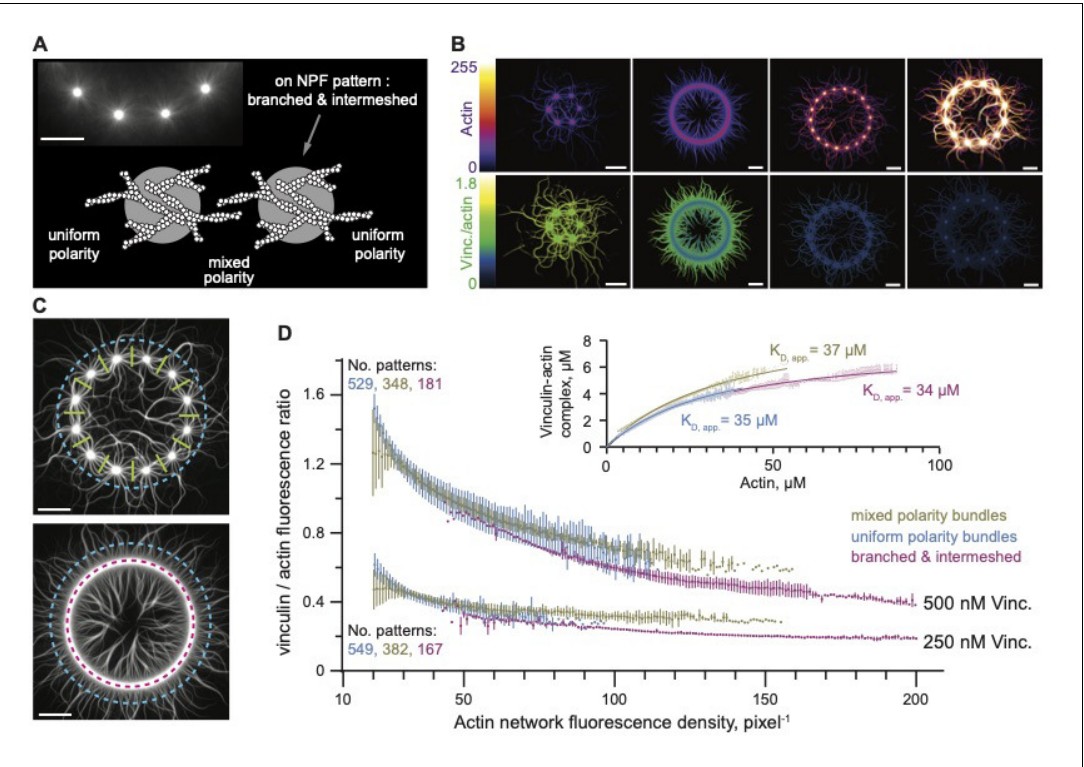

**Figure 2.** The affinity of vinculin to actin in variety of network organizations. (**A**) An example of patterned dots used to reconstitute actin networks (upper left corner). Schematic view shows the organization of actin filaments on two adjacent patterned dots. An Arp 2/3 complex-mediated branched actin network organization assembled on the WASP pWA-coated micropatterns (in gray) with 0.8 µM actin, 2.4 µM profilin, and 130 nM Arp2/3 complex. The Arp2/3 complex is activated by pWA and actin branches grow with their barbed end oriented outwards. (**B**) Representative steady state fluorescence images of reconstituted networks showing actin and the corresponding vinculin to actin fluorescence ratio. The image set illustrates the analysis in D. Conditions as in A but supplemented with VBS1-activated vinculin. (**C**) Dotted and continuous patterned rings indicating the line-scans used for quantifications of actin and vinculin fluorescence intensities across mixed polarity (green), uniform polarity (blue) and branched networks (purple). Conditions as in B. (**D**) Vinculin to actin fluorescence intensity ratio as a function of actin network fluorescence density. Conditions as in B. Inset, Steady state saturation curve was fitted assuming mass-action binding at equilibrium. In all cases, the concentration of vinculin was much lower than the $K_D$ and that of actin was much higher. Apparent dissociation equilibrium constant had a confidence interval of 31 to 38 for uniform polarity, 35 to 39 for mixed polarity, and 31 to 36 for the branched networks. (**D**) Inset. Error bars represent standard deviation from 529, 348, 181 patterns, as indicated (details in Materials and methods section). Scale bars, 20 µm.

The online version of this article includes the following video and figure supplement(s) for figure 2:

**Figure supplement 1.** Overview of tested conditions.
**Figure supplement 2.** Control experiments for activated vinculin binding to higher-order actin structures.
**Figure supplement 3.** Vinculin compacts branched and intermeshed networks.
**Figure 2—video 1.** Assembly of vinculin-free actin networks on patterns.
https://elifesciences.org/articles/53990#fig2video1
**Figure 2—video 2.** Vinculin initiates bundles from uniform- and mixed-polarity actin networks.
https://elifesciences.org/articles/53990#fig2video2
**Figure 2—video 3.** Vinculin initiates bundles from branched and intermeshed actin networks.
https://elifesciences.org/articles/53990#fig2video3

## Rapid reorganization of actin branched networks by vinculin crosslinking

To characterize the effects of vinculin on branched actin networks at the single filament level, we utilized dual-color total internal reflection fluorescence (TIRF) microscopy. Here, we monitored actin

polymerization and vinculin-F-actin interactions simultaneously. Vinculin and Arp2/3 complex were mixed with actin monomers in the absence or in the presence of their activators, VBS1 and the C-terminal amino acids of the WASP-family verprolin-homologous protein (WAVE WA) (*Machesky et al., 1999*), respectively.

Dendritic actin networks were assembled by Arp2/3 complex and rapidly decorated by vinculin, with no detectable lag time between actin polymerization and decoration (*Figure 3A*). Actin branches were initiated on both vinculin-decorated actin filaments and bundles induced by vinculin (*Figure 3A*, *Figure 3—videos 1* and *2*). The branched actin filaments were cross-linked either to the mother filament or to neighboring branches, that is to form uniform- or mixed-polarity bundles, respectively. As polymerization proceeded, the Arp2/3 complex-mediated branched networks were reorganized by vinculin and interconnected via stable and dynamic vinculin-F-actin bundles (*Figure 3—videos 1* and *2*).

Vinculin-decorated actin branches freely diffused or were recruited to existing bundles, with no obvious detectable debranching of the vinculin-decorated network. Quantitative measurements showed that the branch density of dendritic networks was not significantly affected by vinculin decoration, and vinculin binding was not significantly influenced by the presence of Arp2/3 complex (*Figure 3B*). Furthermore, control assays confirmed that the bundling of actin branches was solely due to the presence of activated vinculin (*Figure 3C*) and in the absence of Arp2/3 activator, no

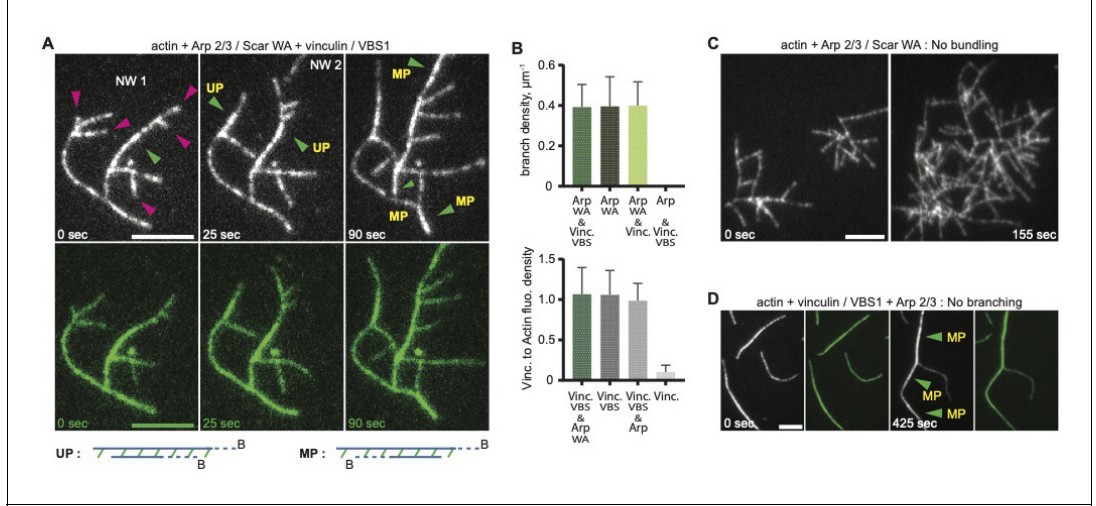

**Figure 3.** Vinculin induced uniform and mixed polarity filaments bundles. (A) Actin polymerization and bundling by 0.3 µM full-length vinculin at, in the presence of 0.9 µM actin, 12 nM Arp2/3 complex, 80 nM WAVE WA fragment, and 1.4 µM talin VBS1. Crosslinking within and between independent networks are followed over time, NW1 and NW2. Dynamic actin branches (purple arrowheads), uniform polarity (UP, green arrowheads) and mixed polarity (MP) bundles are indicated. Schemes defining UP and MP with a pair of actin filaments crosslinked by vinculin (in green) are drawn below; barbed ends (B) are indicated. (B) Quantifications of branch density or vinculin to actin fluorescence ratio of networks polymerized with the mentioned proteins at concentrations indicated in A, are shown as histograms. Error bars represent the standard deviation from 15 to 20 values per condition. (C, D) Controls assays with the mentioned proteins at concentrations indicated in A. Scale bars, 5 µm.

The online version of this article includes the following video and figure supplement(s) for figure 3:

**Figure supplement 1.** Controls for bundling and vinculin integrity after chemical labelling.

**Figure supplement 2.** Vinculin-mediated fast zippering does not affect actin filament dynamics.

**Figure supplement 3.** Actin bundling induced by Vt and full-length vinculin as revealed by cryo-ET.

**Figure supplement 4.** Structural characterization of in vitro vinculin-actin bundles.

**Figure supplement 5.** Directionality analysis of actin filaments.

**Figure supplement 6.** Confidence and neighborhood analyses of actin filaments.

**Figure 3—video 1.** Assembly of branched actin networks without vinculin.

https://elifesciences.org/articles/53990#fig3video1

**Figure 3—video 2.** Vinculin initiates actin bundles from Arp2/3 complex-branched arrays.

https://elifesciences.org/articles/53990#fig3video2

**Figure 3—video 3.** Vinculin-mediated crosslinking can induce bent actin filament breakage.

https://elifesciences.org/articles/53990#fig3video3

actin branching was detected (*Figure 3D*). This suggests that vinculin may directly interact only with a specific cellular sub-population of Arp2/3 complex or additional cellular factors are presumably required (*Chorev et al., 2014*).

To characterize the bundling activity of full-length vinculin protein in higher resolution and details, we applied hybrid methods combining TIRF microscopy and cryo-ET with an experimental set-up that allowed actin filaments to freely diffuse and adopt random orientations prior to bundling by vinculin. While actin polymerization was not affected by the presence of either VBS1 or vinculin alone and chemical labelling of vinculin did not affect its native auto-inhibited configuration (*Figure 3—figure supplement 1*), assembly kinetics confirmed that actin polymerization was not altered by vinculin binding (*Figure 3—figure supplement 2D*). Here, we calculated an on-rate constant of the actin filament's fast-growing barbed end of $10 \pm 1$ $\mu M^{-1}$ $sec^{-1}$, similar to values obtained for free barbed end actin growth in solution. Our findings were in agreement with Leclainche and collaborators *Le Clainche et al., 2010* who used bacterially expressed full-length vinculin. However, we measured no activation of actin polymerization by vinculin, as was shown by bacterially expressed chicken vinculin tail fragment (*Jannie et al., 2015*; *Wen et al., 2009*). Since we detected no apparent lag time between filament polymerization and vinculin binding (*Figure 3—figure supplement 2A,B*), an on-rate constant of vinculin can be approximated to $10 \pm 1$ $\mu M^{-1}$ $sec^{-1}$, as both proteins were used at similar concentrations. Interestingly, we found that the 'zippering' process was strikingly fast, reaching a velocity of $38 \pm 17$ $\mu m/min$, equivalent to $230 \pm 110$ actin subunits/sec, with zippering angles that exceed 60 degrees (*Figure 3—figure supplement 2C*). These crosslinks were mechanically robust enough to occasionally break existing actin filaments (*Figure 3—video 3*), and the bundles were flexible and could buckle with curvatures exceeding $0.2 \pm 0.06$ $\mu m^{-1}$ for up to 9-filament bundles (*Bathe et al., 2008*; *Figure 3—figure supplement 2E*).

Using cryo-electron tomography (cryo-ET) combined with image processing approaches we resolved the organization and polarity of vinculin-F-actin bundles. Bundles induced by the vinculin tail domain showed an inter-filament spacing, $10 \pm 1$ nm (*Figure 3—figure supplement 3A*), as previously reported (*Janssen et al., 2006*). However, bundles assembled with activated full-length vinculin showed a wider inter-filament spacing, $29 \pm 5$ nm, presumably due to the lack of vinculin heads anchorage through cellular integrin-bound talin (*Case et al., 2015*; *Figure 3—figure supplement 3*). More importantly, we reconstructed actin filaments to a resolution of 14 Angstrom and unambiguously determined filaments' polarity within these in vitro bundles. Next, we determined the polarity of the original filaments (*Figure 3—figure supplement 4A–D* and Materials and methods section, *Martins et al., 2020*). Evaluating the polarity of each actin filament with respect to its neighboring filaments in 3D revealed a mixed-polarity organization, wherein neighboring filaments could exhibit both uniform- and mixed-polarity orientation (*Figure 3—figure supplement 4E*).

Thus, our light microscopy and cryo-ET results suggested that, in a force-independent system, VBS1-activated vinculin mediates stable bundling through a rapid binding and fast crosslinking process. Moreover, this bundling does not favor uniform over a mixed actin polarity for bundled filaments.

## Substrate-bound VBS1-activated vinculin initiates bundles out of branched networks

Cytosolic vinculin is recruited to early integrin adhesions (e.g. nascent adhesions and focal complexes) (*Choi et al., 2008*; *Zaidel-Bar et al., 2003*), and is activated at these adhesion sites by talin, where it is exposed to the shear flow generated by lamellipodial actin retrograde flow (*Case et al., 2015*; *Thievessen et al., 2013*). It is currently unclear how adhesion localized vinculin is involved in local actin bundle formation and how the centripetally flowing F-actin sheet is transformed into a FA-associated bundle.

To address these questions, we spotted talin VBS1 onto micropatterned surfaces before flowing a solution containing soluble, inactive vinculin, WA-activated Arp2/3 complex, and G-actin over the surfaces. We then followed actin dynamics using TIRF microscopy (*Figure 4A*). We observed the rapid recruitment of vinculin to the VBS1-anchored spots (*Figure 4B*, first two panels, and *Figure 4—video 1*). In parallel control experiments, micropatterns coated with bovine serum albumin protein (BSA) did not produce any significant vinculin recruitment (*Figure 4—figure supplement 1*, *Figure 4—video 2*). These results suggest that vinculin was specifically recruited and activated by the immobilized VBS1. Furthermore, control experiments combining microfluidics and TIRF microscopy

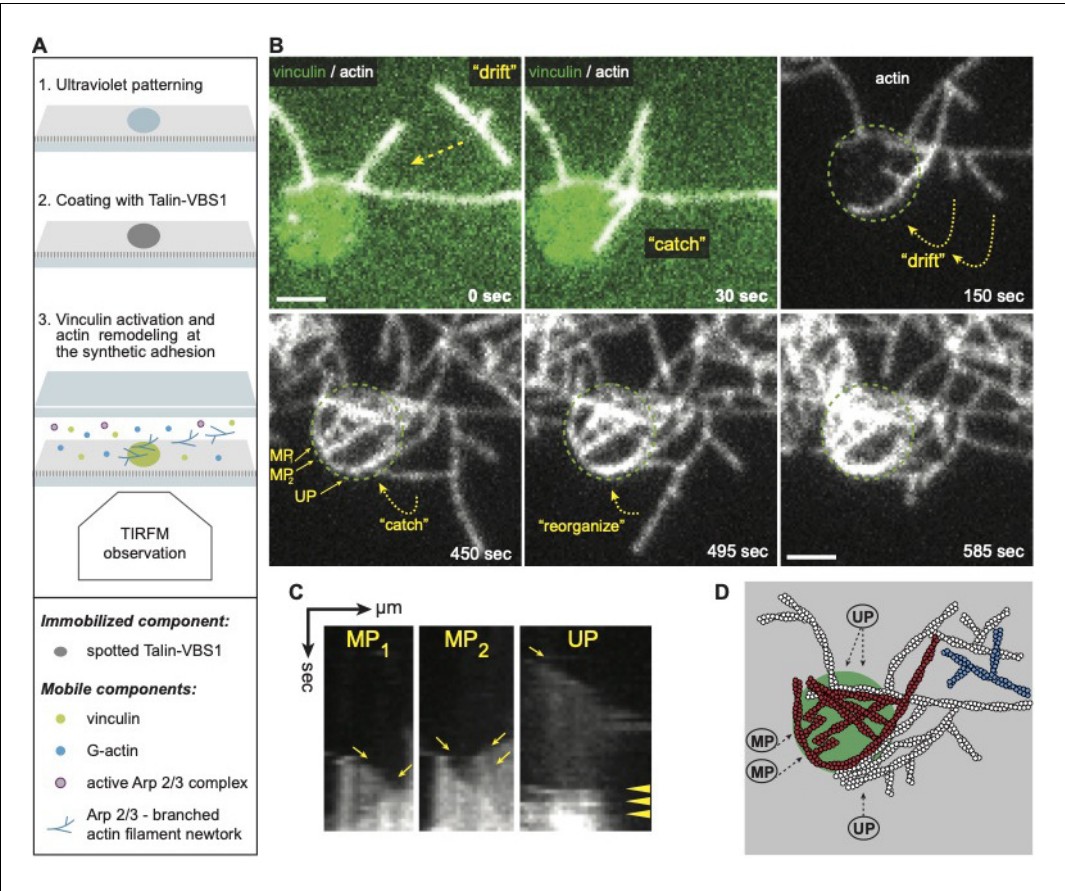

**Figure 4.** Vinculin activated by surface immobilized VBS1 initiate actin bundles only at coated spots. (**A**) Scheme of the experimental setup shows that the passivated coverslip is patterned (1) and coated with talin-VBS1 (2). Next the patterned surface was mounted into a flow chamber in which a mix of 300 nM inactive full-length vinculin, 0.3 µM actin monomer, 18 nM Arp2/3 and its activator, 80 nM WAVE WA, was delivered (3). (**B**) Bundling of Arp2/3-branched actin networks by vinculin was observed only at the sites of patterned VBS1. Dotted arrows indicate the direction of movement of the branched networks, which are bundled. Dotted circle shows the position of the VBS1-coated pattern decorated by vinculin (green in the upper panel). Mixed polarity (MP) and uniform polarity (UP) bundles formed on the pattern. Scale bar, 2 µm. (**C**) Kymographs along the three bundles, shown in B, confirm the crosslinking of Arp2/3 complex-mediated branches into MP (highlighted by arrows) and UP (arrowheads highlight the fusion of 4 branches to a secondary network). (**D**) The schematic view depicts the bundles initiated from the two branched networks, colored in white and in red and imaged in B.

The online version of this article includes the following video and figure supplement(s) for figure 4:

**Figure supplement 1.** Control assay for the specific recruitment and activation of vinculin by surface-bound Talin-VBS1.

**Figure supplement 2.** VBS1-activated Vinculin anchored to actin bundles can engage and bundle undecorated flowing filaments.

**Figure supplement 3.** VBS1-activated Vinculin anchored to actin bundles can engage and bundle undecorated flowing branched networks.

**Figure 4—video 1.** Recruited by VBS1-coated spot, vinculin initiates actin bundles from mobile branched networks.
https://elifesciences.org/articles/53990#fig4video1

**Figure 4—video 2.** Interaction of a mobile branched actin network with BSA-coated spot.
https://elifesciences.org/articles/53990#fig4video2

**Figure 4—video 3.** Immobilized on actin bundles, vinculin can bind mobile actin filaments.
https://elifesciences.org/articles/53990#fig4video3

confirmed that VBS1-activated vinculin immobilized on actin filaments is active and can engage flowing and naked actin filaments (*Figure 4—figure supplements 2* and *3*, *Figure 4—video 3*).

As actin polymerization proceeded, branched actin networks were formed in solution, flowing over the immobilized VBS1-vinculin complexes and eventually being engaged by them. Strikingly, at these functionalized spots, the branched networks were profoundly rearranged into growing uniform- and mixed-polarity bundles (*Figure 4B–D*, *Figure 4—video 1*). Moreover, the actin bundling process was strictly local, occurring only at the immobilized VBS1-vinculin spots, but not observed elsewhere. In the control experiments, BSA-coated patterns showed no apparent interactions and no network remodeling of actin was detected (*Figure 4—figure supplement 1*, *Figure 4—video 2*). The fact that vinculin, once selectively recruited and activated by spotted talin VBS1, is able to bind and bundle branched actin networks is highly reminiscent of actin bundling along focal complexes (*Bachir et al., 2014*; *Case et al., 2015*; *Choi et al., 2008*; *Thievessen et al., 2013*) and maturing FAs (*Alexandrova et al., 2008*; *Gardel et al., 2008*; *Geiger et al., 2009*; *Tee et al., 2015*).

## Discussion

Vinculin is a canonical component of the integrin-mediated adhesion machinery (*Geiger, 1979*). It interacts with multiple adhesome proteins (*Horton et al., 2016*; *Zaidel-Bar et al., 2007a*; *Zamir and Geiger, 2001*) and therefore participates in interactions between the ECM and the actin cytoskeleton. Previously, it was shown that vinculin exists in an auto-inhibited state (*Bakolitsa et al., 2004*; *Borgon et al., 2004*; *Izard et al., 2004*). Vinculin is activated after binding to talin at nascent adhesion sites and subsequent exposure to mechanical force, driving focal adhesion assembly (*Galbraith et al., 2002*; *Thievessen et al., 2013*). Monitoring the early stages of adhesion formation using live cell microscopy demonstrated that the assembly of primordial adhesions ('nascent adhesions' and 'focal complexes') at the rear of the lamellipodium is followed by centripetal extension of actin filaments from these sites (*Bachir et al., 2014*; *Choi et al., 2008*; *Humphries et al., 2007*; *Tee et al., 2015*; *Thievessen et al., 2013*). Understanding the properties of proteins that participate in the initiation and maturation of these nascent adhesion-cytoskeleton interactions is of a major importance.

Inspired by the dynamic interplay between the branched actin network, generated at leading edge, that flows centripetally, and the subjacent, nascent integrin adhesions (*Case and Waterman, 2015*; *Geiger et al., 2009*), we searched for an in vitro 'molecular model' that may simulate early events of adhesion assembly (*Choi et al., 2008*; *Thievessen et al., 2013*). In pursuit of this goal, we focused on the interplay between relevant cytoskeletal network components and full-length vinculin. A combination of real-time fluorescence microscopy, microfluidics, advanced substrate micro-patterning and cryo-ET provided insights into the dynamic molecular processes whereby active, nascent adhesion-associated vinculin transforms Arp2/3-induced branched actin networks into mature adhesion-associated bundles.

Relatively little is known about how vinculin controls the organization of lamellipodial branched actin networks, despite its crucial role in bridging actin dynamics and adhesions turnover. In vivo high-resolution traction-force and fluorescence microscopy showed that the vinculin tail domain mediates interactions of F-actin retrograde flow to the mature focal adhesion (*Alexandrova et al., 2008*; *Thievessen et al., 2013*), and plays a critical role controlling nascent adhesion maturation (*Choi et al., 2008*; *Thievessen et al., 2013*) and organization of the lamellipodial-lamellar actin border (*Alexandrova et al., 2008*; *Thievessen et al., 2013*). However, the underlying molecular mechanisms remain unclear. Here we showed that the vinculin-binding domain of Talin1 (VBS1) activates full-length vinculin, enabling it to interact and control the dynamics and the organization of reconstituted lamellipodium-like Arp2/3 complex-mediated actin networks (*Figure 1*). Our reconstitution elucidates a fundamental standing question of how vinculin, activated in a force-independent regime, induces a mesoscale reorganization of lamellipodial branched actin networks.

Our results show that, while binding talin VBS1, vinculin readily decorated and reorganized Arp2/3 complex-mediated branched networks of variable density and stiffness (*Figures 2* and *3*). The interaction between vinculin and F-actin was fast and independent of actin polymerization resulting into the formation of mixed-polarity bundles. Yet, it is worth noting that in the in vitro experiments (*Figure 3—figure supplement 3*) the full-length vinculin protein may not fit within the tight actin

bundle that is formed by vinculin tail domain alone (*Janssen et al., 2006*). In cells, vinculin binds to integrin-anchored talin, therefore 3D actin bundles crosslinked only by vinculin are unlikely.

Previous structural investigations (*Bakolitsa et al., 2004*; *Borgon et al., 2004*; *Dedden et al., 2019*; *Goult et al., 2013*; *Izard et al., 2004*), in vitro stretching assays (*del Rio et al., 2009*; *Yao et al., 2015*), live cell microscopy studies (*Margadant et al., 2011*; *Thievessen et al., 2013*), and Förster resonance energy-based tension sensors for both proteins (*Case et al., 2015*; *Grashoff et al., 2010*; *Kumar et al., 2016*) have revealed the mechanosensitive nature of talin and vinculin. These studies support the current model requiring actin-generated force to unveil key regulatory domains that potentiate their role in adhesion initiation and maturation. However, recent studies propose an alternative pathway to the force-dependent relief of auto-inhibition in talin and vinculin. Vinculin binding site R3 (*Izard et al., 2004*) was found to be available for vinculin binding in soluble closed talin (*Dedden et al., 2019*), while Paxillin was shown to promote efficient targeting of vinculin to the adhesion site prior to its activation by talin (*Case et al., 2015*). Recent in vivo reconstitutions confirmed the key role of R3 in the binding of vinculin to talin in a force-independent regime, as well as paxillin association to closed vinculin and talin (*Atherton et al., 2020*). These interactions may form precomplexes at an early-stage nascent adhesion (*Choi et al., 2008*). Furthermore, in vivo cross-correlated fluctuations analysis producing high-resolution spatial and temporal maps (*Digman et al., 2009*) showed that talin-vinculin association precedes their recruitment to nascent adhesions (*Bachir et al., 2014*). Finally, Kelley and colleagues showed that phosphoinositides enable talin-vinculin-actin association in a force independent-regime (*Kelley et al., 2020*). The work presented here supports these findings and shows that vinculin, activated in a force-independent regime, can reorganize Arp2/3 complex dendritic networks.

## Vinculin bundling properties

Quantitative and structural analyses of talin VBS1-activated vinculin-induced reorganization of patterned actin structures (*Figure 2*) and single filaments (*Figure 3*) revealed fundamental features of vinculin-actin interactions: (i) activated vinculin binds to uniform, mixed-polarity and intermeshed branched actin organizations with a similar affinity; (ii) three-dimensional bundles assembled were characterized by a random polarity neighborhood; (iii) activated vinculin displayed an association rate constant resembling that of actin polymerization and filaments zippering rate of ~0.6 μm/sec. This rate is considerably faster than the lamellipodial actin polymerization rate, which falls in the range of 1–11 μm/min (*Geraldo et al., 2008*; *Medeiros et al., 2006*; *Watanabe and Mitchison, 2002*). It infers that the zippering is independent of actin polymerization, suggesting that vinculin-mediated crosslinking is not a rate-limiting step at the leading edge of migrating cells, where nascent adhesions emerge under the highly dynamic lamellipodial actin network (*Choi et al., 2008*; *Ponti et al., 2004*; *Zaidel-Bar et al., 2003*). Altogether, our data describes talin-activated vinculin as an efficient candidate which (in the absence of any tensile force) can efficiently bind randomly-oriented actin branches of mobile lamellipodial meshwork (*Svitkina and Borisy, 1999*), forming mixed-polarity bundles, modifying the meshwork organization and halting its dynamics (*Alexandrova et al., 2008*; *Choi et al., 2008*; *Hu et al., 2007*; *Shemesh et al., 2009*). We speculate that in the absence of tensile forces, talin-vinculin precomplexes (*Atherton et al., 2020*; *Bachir et al., 2014*; *Kelley et al., 2020*) efficiently reorganize the mobile lamellipodial actin branches into random polarity bundles, which present at a random orientation (*Svitkina and Borisy, 1999*). This may trigger the formation of a self-sustained system whereby actin retrograde flow exerts force through preformed bundles on talin-vinculin complexes, completing their activation and association to integrins (*Bachir et al., 2014*). This would represent a selective process whereby maturation of surviving nascent adhesions (*Choi et al., 2008*) is regulated by an elegant feedback mechanosensitive mechanism that may reinforce uniform polarity bundles (*Case and Waterman, 2015*; *Huang et al., 2017*).

## Bundled, dendritic networks at talin-bound spots, in absence of tension

Nascent adhesions emerge at the basal aspect of the lamellipodium where they are linked to the extracellular matrix, attenuating the retrograde lamellipodial actin flow without the presence of a local actin bundle (*Gardel et al., 2008*; *Geiger et al., 2009*; *Zaidel-Bar et al., 2007b*; *Zhang et al., 2008*).

Our experiment showed that immobilized talin VBS1 recruits and activates vinculin which initiates linear bundle formation by reorganizing flowing Arp2/3-actin networks (*Figure 4—video 1*). This implies that surface-bound talin-vinculin complex can initiate actin bundles from flowing actin networks and connects them at initial stages of adhesion formation (*Choi et al., 2008*; *Thievessen et al., 2013*). At adhesion sites, such crosslinked actin bundles may be formed by few vinculin proteins occupying several vinculin bindings sites on a talin protein or by vinculin dimers. These hypotheses are summarized in a dynamic 'core mechanism' model for vinculin-actin filament interactions throughout adhesion initiation and maturation (*Figure 5*).

Beside vinculin, other actin crosslinkers, for example α-actinin 1, could concomitantly be recruited and function during the formation of nascent adhesion (*Bachir et al., 2014*; *Otey et al., 1990*). Since α-actinin harbors a cryptic VBS (*Le et al., 2017*) that is likely to be exposed due lamellipodial-generated tension, it may contribute to activating vinculin during adhesion formation. However, α-actinin one may also contribute to crosslinking actin filament bundles anchored by integrin-bound talin-vinculin complexes (*Case and Waterman, 2015*; *Ciobanasu et al., 2014*).

The work described here presents mechanistic insights into the activation-dependent interactions of full-length vinculin with actin networks, crucial for initiation and maturation of nascent adhesions. It provides a molecular view on the mechanical interplay between talin-activated vinculin and the retrograde flow of branched actin networks, independent of myosin-mediated tension. These findings provide mechanistic insights on the force dependent molecular regulation of cell-matrix interactions and cell migration.

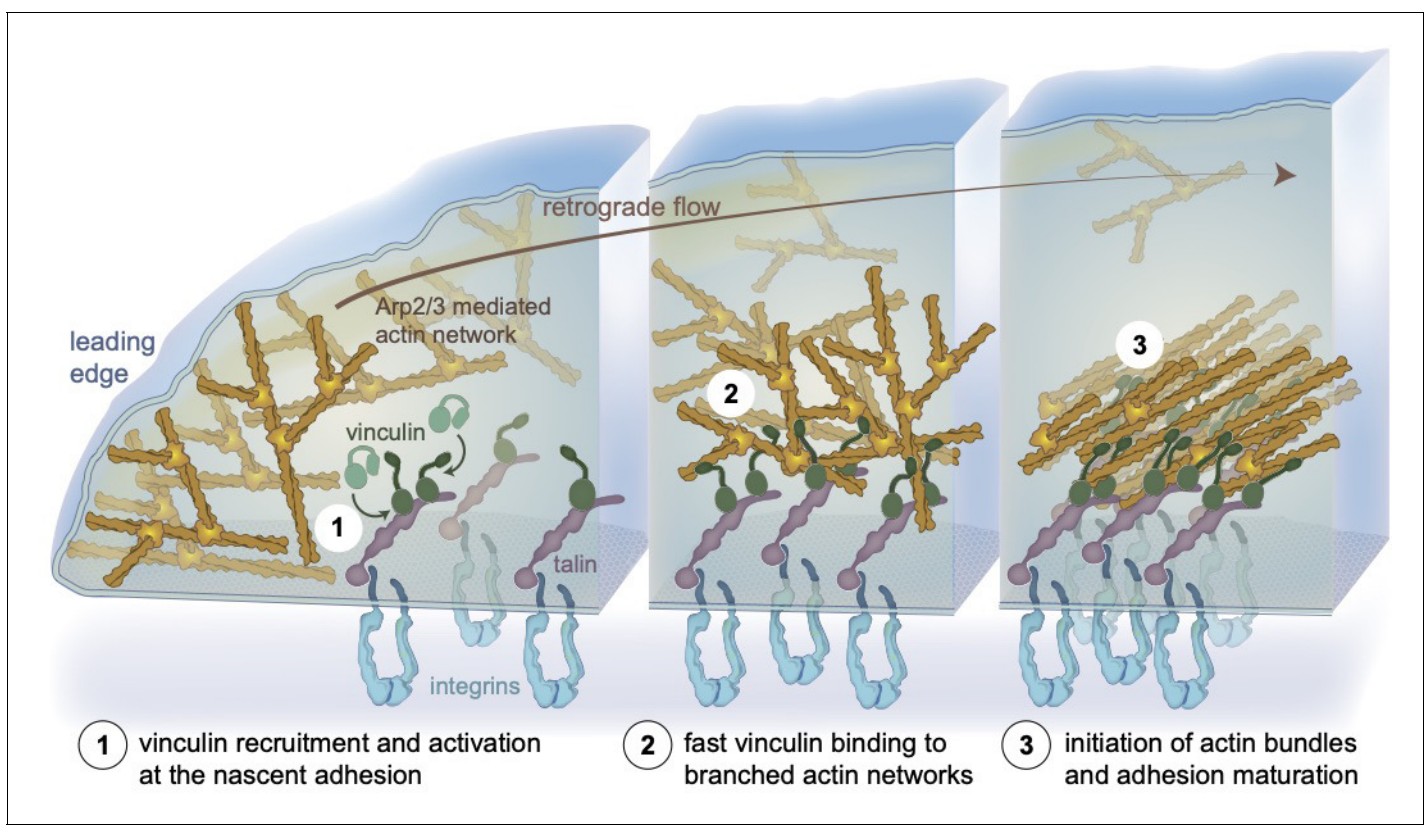

**Figure 5.** A model depicting the initiation of actin bundles from branched networks by membrane bound talin-activated vinculin at nascent adhesions, independent of myosin-mediated force. Schematic representation of a nascent integrin-based adhesion site, localized in proximity to the leading edge of a cell (left). (1) The Arp2/3 branched actin network treadmills and flows centripetally. Vinculin is activated by interacting with talin (*Atherton et al., 2020*; *Kelley et al., 2020*). (2) At nascent adhesion sites, talin-activated vinculin stably binds and bundles mobile branched networks likely affecting actin dynamics. (3) Actin retrograde flow generates tension that further activates talin and vinculin, reinforcing the link to integrins. Adhesion-anchored vinculin interacting with the flowing, branched actin networks initiates bundles at the nascent adhesion site. Vinculin engages the actin retrograde flow, applying tension to the mechanoresponsive components of the adhesion, enabling adhesion self-sustained assembly dynamics and recruitment of additional FA components such as myosin II, α-actinin, and zyxin (*Case et al., 2015*; *Choi et al., 2008*; *Thievessen et al., 2013*).

# Materials and methods

**Key resources table**

| Reagent type (species) or resource | Designation | Source or reference | Identifiers | Additional information |
| --- | --- | --- | --- | --- |
| Gene (*Homo sapiens*) | VCL | Human Genome Nomenclature Database | HGNC:12665 | |
| Cell line used for cloning (*human skin*) | Human A431 | Sigma-Aldrich | Cat# 85090402 | Cell line for cloning VCL |
| Strain, strain background (S. frugiperda) | SF9 cells | ThermoFisher Scientific | Cat# 12659017 | Cell line for baculovirus generation and protein expression |
| Strain, strain background (*E. coli*) | DH5α | ThermoFisher Scientific | Cat# 18265017 | Chemically competent cells |
| Strain, strain background (*E. coli*) | DB3.1 | Gift from Raymond Dutzler | | Chemically competent cells |
| Strain, strain background (*E. coli*) | BL21(DE3)pLysS | Sigma-Aldrich | Cat# 69451 | Chemically competent cells |
| Recombinant DNA reagent | pFBXC3GH-VCL | This paper, Subcloned in vector gifted by R. Dutzler | | For recombinant baculovirus generation and full-length human VCL expression in SF9 cells |
| Recombinant DNA reagent | pBXNHG3-TLN1-VBS1 | This paper, Subcloned from gift from Christophe LeClainche, in vector gifted by R. Dutzler | | Human Talin1-VBS1 (482–636) |
| Recombinant DNA reagent | pBXNHG3-VCL-Tail | This paper, Subcloned from pFBXC3GH-VCL | | Human vinculin tail (879–1066) |
| Commercial assay, kit | Bac-to-Bac Vector Kit | ThermoFisher Scientific | Cat# 10359016 | For cloning |
| Commercial assay, kit | RNeasy mini Kit | Qiagen | Cat# 74104 | For cloning |
| Commercial assay, kit | AffinityScript qPCR cDNA Synthesis Kit | Agilent | Cat# 600559 | For cloning |
| Chemical compound, drug | Alexa Fluor 488 C5 Maleimide | ThermoFisher Scientific | Cat# A10254 | For protein labeling |
| Chemical compound, drug | Alexa Fluor 647 C2 Maleimide | ThermoFisher Scientific | Cat# A20347 | For protein labeling |
| Chemical compound, drug | Alexa Fluor 568 C5 Maleimide | ThermoFisher Scientific | Cat# A20341 | For protein labeling |
| Chemical compound, drug | mPEG-Silane, MW 5 k | Creative PEGWorks | Cat# PLS-2011 | Surface passivation |
| Chemical compound, drug | PLL(20)-g(3.5)-PEG(2) | SuSoS AG | | Surface passivation |
| Chemical compound, drug | Methyl Cellulose | Sigma-Aldrich | Cat# M0387 | Reconstitution assays |
| Peptide, recombinant protein | Glucose Oxidase Type II | Sigma-Aldrich | Cat# G6125 | Reconstitution assays |
| Peptide, recombinant protein | Catalase | Sigma-Aldrich | Cat# C9322 | Reconstitution assays |
| Other | Adhesive Tape | Nitto | Cat# 5601 | Microscopy chamber assembly |
| Other | sticky-Slide VI 0.4 | Ibidi | Cat# 80608 | Reconstitution assays |

*Continued on next page*

*Continued*

| Reagent type (species) or resource | Designation | Source or reference | Identifiers | Additional information |
|---|---|---|---|---|
| Software, algorithm | FIJI | | https://imagej.net/Fiji/Downloads | For data analysis |
| Software, algorithm | Prism | GraphPad | https://www.graphpad.com/scientific-software/prism/ | For data analysis |
| Software, algorithm | MATLAB | MathWorks | https://www.mathworks.com/products/matlab.html | For data analysis |
| Software, algorithm | SerialEM | University of Colorado Open source under an MIT license | https://bio3d.colorado.edu/SerialEM/download.html | For data acquisition |
| Software, algorithm | UCSF Chimera | University of California | https://www.cgl.ucsf.edu/chimera/download.html | For structure visualization |

## Human full-length vinculin cloning

Full-length vinculin cDNA was isolated from a human A431 cell line by extracting mRNAs using the RNeasy Kit (QIAGEN), followed by a reverse transcriptase reaction with a specific primer (5'-GG TGCCTACTGGTACCAGGGAG-3') using the AffinityScript kit (Agilent). The PCR product (using the forward 5'-ATATATGCTCTTCTAGTATGCCAGTGTTTC-3', and the reverse 5'-TATATAGCTCTTCA TGCCTGGTACCAGGG-3' primers) of 1066 amino acids long, was cloned into a pFBXC3GH insect cell expression vector containing a C-terminal 3C protease cleavage site followed by a GFP and His tag, using the fragment exchange (FX) cloning strategy (*Geertsma and Dutzler, 2011*). Protein expression was performed by generating a recombinant baculovirus for Sf9 insect cell infection at a density of $2.0 \times 10^6$ mL$^{-1}$ using the Bac-to-Bac system (Invitrogen). Briefly, the recombinant vinculin-GFP encoding plasmid pFBMS was transformed into DH10Bac *E. coli* cells, which enabled the transposition of the recombinant gene into the bacmid genome. The recombinant bacmid DNA was then isolated and transfected into Sf9 cells to generate the full-length vinculin-expressing baculovirus.

## Human full-length vinculin tail (879–1066) cloning

Human vinculin tail (879–1066) was cloned from pFBXC3GH-human FL vinculin into a pBXNHG3 bacterial expression vector, containing an N-terminal His tag followed a GFP and 3C protease cleavage site, using the fragment exchange (FX) cloning strategy (*Geertsma and Dutzler, 2011*).

## Protein expression and purification

Sf9 cells were infected with the full-length vinculin-expressing baculovirus for recombinant protein expression. Expressing cells were harvested by gentle centrifugation and lysed by sonication in 20 mM Tris-HCl pH 7.5, 0.4 M NaCl, 5% Glycerol, 1 mM DTT, 0.1% Triton, and protease inhibitors. Full-length vinculin or vinculin tail fragment 879–1066 were purified from clarified cell extract on Profinity Ni-charged Immobilized metal affinity chromatography, IMAC resin (Bio-Rad Laboratories, Inc). The C-terminal GFP followed by a 10xHis tag was cleaved by incubating the resin-bound protein with 3C protease, and full-length vinculin was eluted with 20 mM Tris Ph 7.5, 150 mM NaCl, 5% glycerol, 3 mM beta mercaptoethanol, 5 mM Imidazole. Proteins were further purified on a Superdex 200 size exclusion column (GE Healthcare) and eluted with 20 mM Tris-HCl pH 7.8, 0.15 M KCl, 1 mM MgCl$_2$ and 5% Glycerol. Protein aliquots supplemented with 15–20% glycerol were flash-frozen in liquid nitrogen and stored at −80°C.

Actin was purified from rabbit skeletal muscle acetone powder, purified according to the method of *Spudich and Watt, 1971*, and stored in G-buffer (5 mM Tris-HCl pH 7.8, 0.2 mM CaCl$_2$, 0.5 mM DTT, 0.2 mM ATP). The Arp2/3 complex was purified for bovine thymus by WAVE WA affinity chromatography as described (*Egile et al., 1999*). Human profilin, mouse capping protein (α1/β2), human WAVE WA and human WASP pWA constructs were obtained from L. Blanchoin. Human profilin and His-tagged mouse capping protein constructs were expressed in BL21 (DE3)pLysS strain and purified using ion-exchange and size exclusion chromatography as described (*Lu and Pollard, 2001*; *Palmgren et al., 2001*). GST-tagged human WAVE WA or human WASP pWA fragment constructs, and GST-tagged *S. cerevisiae*. Human Talin 1 VBS1 residues 482–636 and human α-actinin one

constructs were obtained from C. Le Clainche, expressed in BL21 (DE3)pLysS strain and purified by glutathione-Sepharose affinity chromatography as described (*Ciobanasu et al., 2015*). Purified polypeptides were supplemented with 10–20% glycerol, aliquoted, flash-frozen in liquid nitrogen and stored at −80°C. Ca-monomeric actin was stored on ice and used within 2 weeks for TIRFM and micropatterning assays.

## Fluorescence labeling of protein

Proteins were labeled through cysteines with Alexa Fluor maleimide dyes (Invitrogen) or Cy3B maleimide dye (GE Healthcare). Actin was labeled with Alexa647, for TIRFM and with Alexa568 for epifluorescence microscopy, vinculin was conjugated to Alexa488, α-actinin one to Alexa568. Labeling reactions of vinculin or α-actinin one were performed in 50 mM K-Phosphate buffer pH 7, and actin labeling in 2 mM Tris-HCl pH 7, 0.2 mM $CaCl_2$, 0.2 mM ATP, for 16 hr. To limit excessive labeling, protein/dye molar ratio was kept ≤1:3. The reactions were stopped by adding 10 mM DTT. The labeled vinculin was purified using Superdex 200 size exclusion column (GE Healthcare), in 20 mM Tris-HCl pH 7.8, 0.15 M KCl, 1 mM $MgCl_2$ and 5% Glycerol. Labeled actin was polymerized, depolymerized, and gel-filtered using Superdex 200 size exclusion column (GE Healthcare), in 2 mM Tris-HCl pH 7, 0.2 mM $CaCl_2$, 0.2 mM ATP, 0.5 mM DTT. The degree of labeling (DOL) was determined by absorption spectroscopy, using the known extinction coefficients for the dyes and proteins, or using coefficients predicted from the amino acid sequences using PredictProtein software (ExPASy, Swiss Institute Bioinformatics Resource portal). Actin, vinculin, and α-actinin 1 DOLs ranged between 25–35%. Purified polypeptides were supplemented with 10–20% glycerol, aliquoted, flash-frozen in liquid nitrogen and stored at −80°C. Ca-monomeric actin was stored on ice and used within 2 weeks for TIRFM and micropatterning assays.

## Glass surface passivation and deep-ultraviolet (UV) micropatterning

Using passivated flow cells was a prerequisite for preventing non-specific protein/substrate interactions and improve signal-to-noise ratio during TIRFM image acquisition. Thus, slides and coverslips (CVs) were used to assemble the reaction chambers were drastically cleaned by successive chemical treatments (adapted from *Boujemaa-Paterski et al., 2014*): 2 hr in 2% Hellmanex III, rising in ultrapure water, 30 min in acetone, 30 min ethanol 96%, rinsing in ultrapure water. Slides and CVs were dried using a filtered nitrogen gas flow and oxidized with oxygen plasma (3 min, 30 Watt, Femto low-pressure plasma system Type A, Diener electronic GmbH, Germany), just before an overnight incubation in a solution containing tri-ethoxy-silane-PEG (5 kDa, PLS-2011, Creative PEGWorks, USA) 1 mg/mL in ethanol 96% and 0.02% of HCl, with gentle stirring for TIRFM assays. mPEG-silane coated slides and CVs were then rinsed in ethanol and extensively in ultrapure water. Passivated slides and CVs were then stored in a clean container and used within a week time. Alternatively, for deep UV-assisted micropatterning, plasma cleaned slides and CVs were incubated in PLL(20)-g(3.5)-PEG(2) (SuSoS AG, Switzerland) at 1 mg/mL in Hepes 10 mM pH 7.4. PLL-PEG-coated CVs were immediately patterned, as described below (adapted from *Boujemaa-Paterski et al., 2014*), and mounted onto a PEGylated slide using a double-sided tape (3M electronics or Nitto).

Likewise, passivated flow cells were mounted by gluing a PEG-silane CV onto a PEG-silane slide and used for TIRFM imaging.

## TIRFM imaging

Reconstitution assays were performed using fresh actin polymerization buffer, containing 20 mM Hepes pH 7.0, 40 mM KCl, 1 mM MgCl2, 1 mM EGTA, 100 mM β-mercaptoethanol, 1.2 mM ATP, 20 mM glucose, 40 µg/mL catalase, 100 µg/mL glucose oxidase, and 0.4% methylcellulose. The final actin concentrations were 0.7 to 1.1 µM, with 13–16% Alexa monomer labels. The polymerization medium was supplemented with vinculin, talin VBS1, Arp2/3 complex, WAVE WA, and α-actinin one as indicated in the figure legends and methods (below). The reaction medium was rapidly injected into a passivated flow cell at the onset of actin assembly, imaging started after 2 min. Time-lapse TIRFM was recorded every 5 or 15 s. TIRF images were acquired using a Widefield/TIRF – Leica SR GSD 3D microscope, consisting of an inverted widefield microscope (Leica DMI6000B/AM TIRF MC) equipped with a 160x objective (HCX PL APO for GSD/TIRF, NA 1.43), a Leica SuMo Stage, a PIFOC piezo nanofocusing system (Physik Instrumente, Germany) to minimize the drift for an accurate

imaging, and combined with an Andor iXon Ultra 897 EMCCD camera (Andor, Oxford Instruments). Fluorescent proteins were excited using three solid-state diode lasers, 488 nm (300 mW), 532 nm (500 mW), 642 nm (500 mW). Laser power was set to 5% for Alexa-labeled proteins, and dyes were excited for 50 ms. Image acquisition was performed with 25 degrees-equilibrated samples and microscope stage. The microscope and devices were driven by Leica LAS X software (Leica Microsystems, GmbH, Germany).

## Deep UV-assisted micropatterning and widefield fluorescence microscopy

To direct actin filament polymerization to predefined positions on glass CVs, we used the micropatterning strategy (*Boujemaa-Paterski et al., 2014*; *Reymann et al., 2010*) by printing adhesive patterns on a protein-repellent surface. PLL-PEG-coated CVs were exposed to short-wavelength UV radiation (184.9 nm and 253.7 nm, Jelight, USA) for 2 min through 24 × 24 transparent micropatterns printed on a photomask (Compugraphics, Germany). To ensure high-resolution printing of micropatterns on the PLL-PEG-coated CVs, during UV exposure CVs and the photomask were mounted onto a custom-made vacuum-compatible holder. Immediately after UV exposure, micropatterned CVs were incubated in 0.2–0.3 µM GST-WASp pWA solution for 15 min at room temperature under gentle agitation, then washed in a buffer containing 2 mM Tris-HCl pH 7, 0.2 mM $CaCl_2$, 0.2 mM ATP, 0.5 mM DTT, mounted onto a PLL-PEG coated slide, and injected with a defined reaction medium.

To restrict actin filament polymerization on pWA-functionalization micropatterns, reconstitution media contained both the Arp2/3 complex, recruited and activated by pWA attached to the patterned surfaces, and profilin that inhibits spontaneous nucleation of actin filaments in solution. However, Arp2/3 complex-mediated actin nucleation on the micropatterns is allowed. Thus, to polymerize patterned actin networks, we injected micropatterned flow cells with an actin polymerization medium, containing 20 mM Hepes pH 7.4, 30 mM KCl, 1 mM $MgCl_2$, 1.7 mM EGTA, 40 mM DTT, 1.2 mM ATP, 20 mM glucose, 40 µg/mL catalase, 100 µg/mL glucose oxidase, and 0.2% methylcellulose. In all tests, final actin concentration was 0.8 µM, with 18% Alexa568-labeled monomers, profilin was 2.4 µM, the Arp2/3 complex was 130 µM, vinculin and/or talin VBS1 as indicated in the figure legends. Imaging started after 15 min. Time-lapse microscopy was recorded every 1 or 2 min. To polymerize stiff, dense and motile branched networks, the polymerization medium was supplemented with 16 nM Capping protein, 4 µM actin, with 5% Alexa568-labeled monomers, 8 µM profilin, 120 nM Arp2/3 complex, and vinculin, talin-VBS1 and/or α-actinin 1, as indicated in the figure legends. Fluorescence images were acquired on an inverted widefield Leica DMI4000B microscope equipped with a 63x oil objective (HCX PL APO 63x; NA 1.40–0.60) and combined to Leica DFC 365 FX camera (Leica Microsystems, GmbH, Germany). Illumination was set to 17% of halogen lamp power and fluorescent dyes were excited for 80 ms. Image acquisition was performed with 25 degrees Celsius-equilibrated samples and microscope stage. The microscope and devices were driven by Leica LAS X software (Leica Microsystems, GmbH, Germany). Samples were also explored using an Olympus IXplore SpinSR10 spinning disk confocal imaging system (Olympus Scientific Solutions, USA), equipped with a 60x Silicon oil objective (UPLSAPO UPlan S Apo 60x, NA 1.3) and with two Prime BSI scientific CMOS cameras (Teledyne Photometrics, USA).

## Deep UV-assisted micropatterning and TIRF microscopy

Micropatterns were also coated with GST-talin VBS1 to address vinculin activation to the patterned surfaces. Briefly, immediately after UV exposure, micropatterned CVs were incubated in 0.4 µM GST-talin VBS1 solution for 15 min at room temperature under gentle agitation, then washed in a buffer containing 2 mM Tris-HCl pH 7, 0.2 mM $CaCl_2$, 0.2 mM ATP, 0.5 mM DTT, passivated with a solution of 4.5 µM BSA, washed again, mounted onto a PLL-PEG coated slide, and injected with a defined reaction medium. Control patterns were similarly treated, except for the coating step where GST-talin VBS1 was replaced by a BSA.

To monitor the recruitment of vinculin to the GST-talin VBS1-coated patterns, and subsequent remodeling of Arp2/3-branched actin network to these adhesion-like spots, we injected micropatterned flow cells with an actin polymerization medium, containing 20 mM Hepes pH 7.0, 40 mM KCl, 1 mM MgCl2, 1 mM EGTA, 100 mM β-mercaptoethanol, 1.2 mM ATP, 20 mM glucose, 40 µg/mL

catalase, 100 µg/mL glucose oxidase, 0.4% methylcellulose, 0.2% BSA. Vinculin was 300 nM with 30% Alexa488-labeled monomers, actin was 0.3 µM with 18% Alexa568-labeled monomers, the Arp2/3 complex was 15 nM, WAVE WA was 80 nM. Imaging started when actin filaments enter the evanescent field. Time-lapse microscopy was recorded every 15 s. Control flow chambers assembled with BSA-coated patterns were injected with the same solution. The reaction was monitored with TIRFM using the Widefield/TIRF – Leica SR GSD 3D microscope described above.

## Microfluidics and TIRF microscopy

Microfluidic channels were assembled as described above using a sticky-Slide VI 0.4 (ibidi GmbH, Germany). The microfluidic channel inlet was connected to a PHD 2000Infusion/Withdraw pump (Harvard Apparatus, USA). The flow was set to 25 µL per min. Typically, (1) 0.8–1 µM actin with 18% Alexa568-labeled monomers, 0.5–1 µM vinculin with 30% Alexa488-labeled monomers, 2–4 µM talin VBS1, were supplied in the above-mentioned polymerization buffer from the inlet. (2) At steady state of actin assembly, all soluble components are washed away with five channel volumes, and (3) a solution of a pre-assembled branched actin networks is injected. The branched network was assembled with 0.5 µM actin Alexa647-labeled monomers, 15 nM Arp 2/3 complex and 40 nM WAVE WA, and diluted twofold. Time-lapse imaging of vinculin-actin bundles and branched network started as soon as step (3) was completed. Images were recorded every 15 s with TIRFM using the Widefield/TIRF – Leica SR GSD 3D microscope described above.

## Image processing and data analysis of fluorescence images

Time-lapse videos of filament growth were processed with Fiji software (NIH). For each sample, we measured the rates of barbed end elongation of at least ~10 filaments, typically from at least 20–40 frames of imaging over a span of ~200–400 s, using TIRFM. The growth rate of actin filaments was calculated using Multikymograph Fiji software plugins or a matrix-to-8-bit-image conversion program written in MATLAB, Kymographs were generated over the time course of actin assembly along the actin filament or bundle trace. For filaments or bundles that were stable in the observation field, traces were automatically picked using Multikymograph Fiji software plugins. For mobile structures, traces were manually picked. The alignment of filament pointed-end and internal fluorescence speckles of actin filaments in MATLAB-generated kymograghs accounted for the accuracy of the method.

To measure the zippering velocity of vinculin-induced crosslinking, the length of the zippered region of single filaments was measured over time for in three different experiments where 0.7 to 1.1 µM actin was polymerized in the presence of 600 nM vinculin and 2 µM Talin VBS1, or 300 nM vinculin and 1 µM Talin VBS1, or 300 nM vinculin, 1 µM Talin VBS1, 12 nM Arp2/3 complex and 80 nM WAVE WA. In all three experiments, measured velocities varied within the same range. Determining the number of filaments per vinculin-mediated bundle was conducted by analyzing the fluorescence intensity of bundles using the line-scan tool in Fiji software, and subsequently normalized to the fluorescence intensity of a single actin filament from the same experiment and that was as many times illuminated as the bundle. Additionally, in most cases, time-lapse videos provided bundle formation history hence an evidence from the calculated number of filaments.

To determine the fluorescence ratio of vinculin to actin within patterned actin networks, we analyzed two color widefield fluorescence images taken after 60 min of actin assembly. (1), we piled, separately, the images for each channel (568 for actin or 488 for vinculin signals) and for each patterned geometry, namely continuous circles and dotted circles where the dot-to-dot distance was 6, 11, or 18 µm. (2), Piled images were then aligned according to the patterned geometry using 'Template Matching' plugins of Fiji software. (3), By thresholding the vinculin channel images, we created binary masks to isolate the vinculin-decorated network and filter out the background fluorescence for each channel (568 for actin or 488 for vinculin signals). (4) We then generated ratiometric images (488 fluorescence intensity/568 fluorescence intensity, as shown in *Figure 2B*), using the default plugins in Fiji. (5) Since we kept constant all illumination parameters and all biochemical parameters of the reconstituted assays (except vinculin concentration that was 0.25 or 0.5 µM), we were able to perform quantitative analyses. For all patterns (>1000), the fluorescence density of actin (from actin fluorescence images) and related vinculin to actin fluorescence ratio (from ratiometric images) were measured for each network along the pixel selections presented in *Figure 2C*. For data analyses, we

wrote a MATLAB program to correlate, for each pixel, the vinculin/actin fluorescence ratio to the actin fluorescence density. For each series of pattern geometry (repeated 29 to 70 times), we computed the distribution and the mean value of each network (branched, uniform or mixed polarity) separately. In final statistics, we characterized each type of network by pooling the mean ratio values related to the same actin density value, and calculated their mean and standard deviation, which were plotted as a function of actin fluorescence density (*Figure 2D*). To estimate the apparent equilibrium dissociation constant of vinculin to actin networks, we calibrated the fluorescence intensities to the protein concentration using labelled actin or labelled vinculin. Mean fluorescence intensities for 3–5 known protein concentrations were measured and averaged. Calibrations were carried out using pegylated glass and coverslips using the same reaction chamber, volumes and composition as for the experiments. The mean fluorescence intensities were converted into protein concentration in the networks. This allowed to determine a steady state saturation curve showing local concentration of bound vinculin, which represents the local concentration vinculin-actin complex, as a function of local actin concentration. The apparent equilibrium dissociation constant was then obtained by fitting the steady state saturation curve using Graphpad software and assuming mass-action binding at equilibrium.

## Synthesis of tiopronin Monolayer-Protected AuNPs

Water-soluble tiopronin-protected gold nanoparticles (AuNP-TP) were synthesized as described previously (*Dahan et al., 2018*). Briefly, 0.1 g of Gold(III) chloride trihydrate (Sigma–Aldrich) and 0.11 g of N-(2-mercaptopropionyl) glycine (Tiopronin, Cayman Chemical) were dissolved in 13 mL methanol:acetic acid solution (6:1 v/v) and gently stirred at room temperature for 1 hr. 1 mL of 1M sodium borohydride (Acros Organics) in deionized water was prepared and immediately added to the solution. The solution was vigorously stirred for 2 hr at room temperature. The solution was dialyzed against deionized water for 72 hr with a 8–10 kDa Float-a-Lyzer G2 (Spectrum). Dialyzed solution was stored at 4˚C.

## Functionalization of tiopronin Monolayer-Protected AuNPs

AuNP-TPs were coupled to GST-VBS1 in a two-step reaction. First, a concentrated solution of AuNP-TPs (OD = 1 at 520 nm) in 0.1M, pH6 MES buffer were 'activated' by incubating with 40 mM EDC (1-ethyl-3-[3-dimethylaminopropyl] carbodiimide hydrochloride, Sigma–Aldrich) and 80 mM sulfo-NHS (N-hydroxysulfosuccinimide, Thermo) in 0.1M pH6 MES buffer for 15 min at room temperature. The volume was increased to 1 mL with 0.1M pH6 MES buffer and concentrated using a 30 kDA MWCO centricon (Millipore). This was repeated twice with 0.1M pH6 MES and twice with 2 mM pH 7.4 sodium phosphate buffer. The concentration of the AuNPs was adjusted to an OD = 0.1 at 520 nm and reacted with 1.5 nmole VBS1-GST in a 50 µL final volume for 2 hr at room temperature. The reaction mixture was stored on ice at 4˚C for later use. The conjugation reaction was analyzed agarose gel electrophoresis. 16 µL of sample was mixed with 4 µL 30% glycerol and loaded into a 0.75% agarose gel. The gel was run at 90V for 20 min.

## Sample preparation for cryo-electron tomography

In order to favor a random orientation of filaments in solution, and prevent a biased orientation due to excessive filament length, we incubated short preformed actin filaments with vinculin in the presence of talin VBS1. In a buffer containing 20 mM Hepes, pH 7.0, 50 mM CaCl$_2$, 1 mM MgCl$_2$, 1 mM EGTA, 0.5 mM DTT, 0.2 mM ATP, 10 µM Mg-ATP actin monomers were preassembled in the presence 50 nM capping protein α1β2. Thus, the ratio CP:actin monomers was 1:200 yielding filaments of ~0.7 µm in average, unable of end-to-end annealing. Subsequently, 0.5 µM capped actin filaments were incubated for 30 min at 25˚C with 1.2 µM full-length vinculin in the presence of 3 µM talin VBS1. When GFP-VBS1 was used, 0.3 µM capped actin filaments were incubated for 1 hr at 25˚C with 2 µM vinculin and 8 µM GFP-VBS1. Gold nanoparticles labeling was conducted as followed: 0.3 µM capped actin filaments were incubated for 1 hr at 25˚C with 2 µM vinculin, 0.8 µM AuNP-GST-VBS1, and 7.2 µM GST-VBS1. For the experiments using vinculin tail fragment, 0.3 µM capped actin filaments were incubated for 30 min at 25˚C with 2.2 µM tail fragment. Next, 4 µL of the reconstituted solution and 1 µL of a 10 nm fiducial gold marker solution (Aurion, Netherlands) was applied onto glow-discharged EM grids coated with a holey carbon mesh (Cu R2/1, 200 mesh, Quantifoil).

Finally, the EM grids were manually blotted and plunge frozen in liquid ethane. The EM grids were stored in liquid $N_2$ until used for cryo-ET data acquisition.

## Cryo-ET data collection

Cryo-ET data was acquired on a FEI Titan Krios transmission electron microscope (FEI, Hillsboro, USA) equipped with a GIF Quantum energy filter and a K2-Summit direct electron detector (Gatan, Pleasanton, USA). The microscope was operated at 300 keV in zero-loss mode with the slit width of the energy filter set to 20 eV.

Using SerialEM (*Mastronarde, 2005*), image stacks were recorded at each tilt angle in super-resolution mode with an electron flux of ~8 electrons per pixel per second.

The image stacks were acquired at a dose-fractionated frame rate of 1 frame per 0.2 s at a magnification of 42,000×, resulting in a pixel size of 0.17 nm. The tilt-series covered an angular range of −60° to + 60° and were recorded at increments of 2°, at a defocus of −4 μm and a total accumulated electron dose of ~70 electrons per $Å^2$.

The image stacks were 2 × 2 down-sampled and subjected to motion-correction using Motion-Corr (*Li et al., 2013*), resulting in a final pixel size of 3.4 Å. Next, the tilt-series were contrast transfer function corrected (*Eibauer et al., 2012*) and finally reconstructed in volumes with a size of 1024 × 1024×512 voxel (total down-sampling is 8 × 8, voxel size of 13.6 Å) using the TOM Toolbox (*Nickell et al., 2005*).

Data acquisition for samples containGFP-VBS1 and AuNP-TPs-GST-VBS1 vinculin bundles were acquired using a dose-fractionated frame rate of 6 frames per 1.2 s with a magnification of 64,000 × resulting in a pixel size of 0.11 nm. The tilt-series covered an angular range of − 60° to + 60° and were recorded at increments of 3°, at a defocus of −4 μm and a total accumulated electron dose of ~85 electrons per $Å^2$. Reconstruction of the tomograms was done with IMOD (*Kremer et al., 1996*), using gold fiducials for tilt-series alignment and weighted back-projection reconstructions. Cryo-tomograms were 2 × 2 down-sampled for reconstruction resulting in a pixel size of 2.22 Å.

## Structural analysis of actin filaments from bundles by cryo-ET

Actin filaments were automatically segmented, as we described in *Martins et al., 2020*, from 16 tomograms using a convolutional neural network algorithm (*Chen et al., 2017*). Based on the segmentations, 3D coordinates of actin segments were chosen along the filaments with a spacing of 16.5 nm. Next, actin segments were reconstructed into sub-tomograms of 144 × 144×144 voxels, corresponding to a box size of ~50 nm and pixel size of 3.4 Å. A total number of 39,231 actin segments were extracted. The 3D coordinate extraction from segmentations was performed using MAT-LAB (Mathworks, Natick) and the sub-tomogram reconstruction was performed using the TOM Toolbox.

Next, the sub-tomograms were aligned parallel to the x-axis. A 22 nm thick central section of the aligned sub-tomograms was projected and an 11 nm thick filamentous mask along the x-axis was applied.

Subsequently, we performed one round of 2D classification with the pre-aligned using RELION (*Scheres, 2012*). Thereby, we removed low-quality particles and false positive particles (e.g. gold markers, background), resulting in a cleaned particle set of 29,643 actin segments. Representative 2D classes are shown in *Figure 3—figure supplement 4B*.

Next, we subjected these particles to a helical 3D-refinement using RELION, using a cylinder as an initial template (*He and Scheres, 2017*), a helical rise of 27.6 Å and a helical twist of 166.7 degrees (*Galkin et al., 2015*). The obtained 3D reconstruction of actin (*Figure 3—figure supplement 4C*) resolved to ~14 Å, as indicated by the RELION software. Actin structures were visualized with UCSF Chimera (*Pettersen et al., 2004*).

## Mapping the polarity in actin bundles

Using the segments that contributed to the final actin structure, all transformations that were applied to the segments during reconstruction (that are pre-alignment, projection, and helical reconstruction), were inverted. The inverted transformations describe how an actin segment is oriented at the position of its extraction in the tomograms. In particular, the first Euler rotation of a segment

around the z-axis (termed psi angle) resolves the segments polarity, given that the barbed- and pointed ends can be identified in the reconstructed structure. As expected, histograms of the psi angles show two peaks per bundle, 180° apart (*Figure 3—figure supplement 5*).

In the next step, we connected the segments back to filaments. In principle, this information is contained in the initial segmentations. However, the automatic segmentation procedure tends to fuse filaments, if they run parallel or cross each other with small distances. Therefore, we manually combined the segments into filaments. When filament continuity within the tomogram was questionable, we kept the smaller pieces, so that an unambiguous assignment was possible. Therefore, this procedure underestimates the length of certain filaments. However, it has no influence on the parameter we investigated in this work, namely the polarity and distances between actin filaments.

In order to determine the polarity of filaments we plot the segments belonging to each filament as columns of circles (*Figure 3—figure supplement 5*). The color was assigned according to segment transformation, left (blue) or right (red) psi angle. The columns show a clear tendency to be either blue or red, confirming that we determined the direction of actin segments belonging to the same filaments in a consistent manner. Since the procedure did not consider which segment belongs to which filament, this result serves as an internal control. Since an actin filament can only have a single polarity, we decided the final direction of a filament based on the majority of its segments and mapped the direction of a filament accordingly (*Figure 3—figure supplement 4D*, *Figure 3—figure supplement 5*).

A confidence score measurement providing an additional confirmation to the reliability of determining the polarity of each filament (*Figure 3—figure supplement 6A*). We termed this score the 'majority confidence score'. It is defined as the maximum number of segments pointing in a similar direction divided by the total number of segments of the respective filament. In this study we extracted 606 filaments from 16 tomograms. Around 20% of the filaments had a majority confidence score <2/3 (*Figure 3—figure supplement 6A*). These filaments were excluded from further analysis, for example the black filaments in *Figure 3—figure supplement 4D*. In order to measure the distance between the filaments and to quantify the distribution of uniform polarity and mixed polarity regions within the bundles, we performed a 3D neighborhood analysis that described in details in *Martins et al., 2020*.

In brief, for every actin segment we extracted the polarities of the (up to) three nearest segments of neighboring filaments, within a 3D distance of 40 nm, defining its neighborhood. Based on that we determined for each segment the degree of uniform polarity (UP) and mixed polarity (MP) of its neighborhood (*Figure 3—figure supplement 6C*). Next, we defined a score (MPs-UPs)/(MPs+UPs) to characterize the overall polarity of each bundle.

The calculated scores range from −1 to 1, for example if all filaments in a bundle point in the same direction the score is −1 (*Figure 3—figure supplement 4E*). The 3D neighborhood analysis was programmed in MATLAB.

## Acknowledgements

This study was supported by the Swiss National Foundation, grant no. 31003A_179418, and the Mäxi Foundation (to OM), the Israel Science Foundation (to BG) and the Klaus Tschira Foundation (to BG). The authors would like to thank the Center of Microscopy and Image Analysis of the University of Zürich, and Sascha Weidner from the Workshop of the Department of Biochemistry, the University of Zürich, for the design and production of the mask holder used for the micropatterning. The authors are grateful to Prof. Hans-Werner Fink, Prof. Jürg Osterwalder and Prof. Marta Gibert from the Physik-Institut of the University of Zürich for their help with the UVO cleaner. We thank Ed Egelman (University of Virginia) for fruitful discussions during the initial stages of the project. We thank Julien Berro (Yale University) for insightful discussions. BG is the incumbent of the Erwin Neter Professorial Chair in Cell and Tumor Biology. We thank Dr Agnieszka Kawska (IllusScientia) for the artwork in *Figure 5*.

## Additional information

### Funding

| Funder | Grant reference number | Author |
|---|---|---|
| Schweizerischer Nationalfonds zur Förderung der Wissenschaftlichen Forschung | 31003A_179418 | Ohad Medalia |
| Maxi Foundation | LMAN | Ohad Medalia |
| Israel Science Foundation | | Benjamin Geiger |
| Klaus Tschira Foundation | | Benjamin Geiger |

The funders had no role in study design, data collection and interpretation, or the decision to submit the work for publication.

### Author contributions

Rajaa Boujemaa-Paterski, Conceptualization, Formal analysis, Validation, Investigation, Visualization, Writing - original draft, Writing - review and editing; Bruno Martins, Investigation, Methodology; Matthias Eibauer, Formal analysis, Methodology, Writing - review and editing; Charlie T Beales, Investigation, Writing - review and editing; Benjamin Geiger, Conceptualization, Supervision, Writing - review and editing; Ohad Medalia, Conceptualization, Supervision, Funding acquisition, Writing - original draft, Project administration, Writing - review and editing

### Author ORCIDs

Rajaa Boujemaa-Paterski https://orcid.org/0000-0001-9645-387X
Ohad Medalia https://orcid.org/0000-0003-0994-2937

### Decision letter and Author response

Decision letter https://doi.org/10.7554/eLife.53990.sa1
Author response https://doi.org/10.7554/eLife.53990.sa2

## Additional files

### Supplementary files

• Transparent reporting form

### Data availability

All data generated or analysed during this study are included in the manuscript and supporting files. The structural data is deposited to the EMD, access number EMD-10737, and two tomographic volumes were deposited in EMPIAR under the accession number EMPIAR-10548.

The following datasets were generated:

| Author(s) | Year | Dataset title | Dataset URL | Database and Identifier |
|---|---|---|---|---|
| Boujemaa-Paterski R, Martins B, Eibauer M, Geiger B, Medalia O | 2020 | Actin filament structure from vinculin-induced bundles | https://www.emdatare-source.org/EMD-10737 | EMDataResource, EMD-10737 |
| Boujemaa-Paterski R, Martins B, Eibauer M, Beales C, Geiger B, Medalia O | 2020 | Talin-activated vinculin interacts with branched actin networks to initiate bundles | https://www.ebi.ac.uk/pdbe/emdb/empiar/entry/10548/ | Electron Microscopy Public Image Archive, EMPIAR-10548 |

The following previously published dataset was used:

| Author(s) | Year | Dataset title | Dataset URL | Database and Identifier |
|---|---|---|---|---|
| Galkin VE, Orlova A, Vos MR, Schroder GF, Egelman EH | 2015 | Near-Atomic Resolution for One State of F-Actin | https://www.ebi.ac.uk/pdbe/entry/pdb/3j8i | EMDataResource, EMD-6179 |

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
