## [Decision Letter]

[Editors’ note: the authors submitted for reconsideration following the decision after peer review. What follows is the decision letter after the first round of review.]

Thank you for submitting your work entitled "Talin-activated vinculin interacts with branched actin networks to initiate bundles" for consideration by *eLife*. Your article has been reviewed by four peer reviewers, one of whom is a member of our Board of Reviewing Editors, and the evaluation has been overseen by a Senior Editor. The following individual involved in review of your submission has agreed to reveal their identity: Christoph Ballestrem (Reviewer #4).

Our decision has been reached after consultation between the reviewers. Based on these discussions and the individual reviews below, we regret to inform you that your work will not be considered further for publication in *eLife*.

Overall, there were concerns about the novelty of the work and the new insights that might be gained into function. While some of the points raised below might be addressed with new experiments or even more detailed explanations, it was felt that the scope of the revisions needed were outside of the bounds of what is required for an *eLife* resubmission. We realize that this decision will be disappointing but hope that you will find these reviews helpful in planning your next experiments.

Reviewer #1:

This is a rather complicated study that aims to provide new insights into how talin and vinculin interact with actin to form bundles in cells. The work done is purely in vitro, which is necessary to simplify and control the system. While some aspects appear novel, I will need to defer to other reviewers in terms of the biological significance of the results. I am not bothered at all about the seeming conflicts with Janssen et al., 2006, since Kim et al., 2016, came to very different conclusions. The work on actin polarity appears to have been done well and is convincing.

Reviewer #2:

The finding that vinculin does not decorate actin without vinculin activation is not novel and thus not surprising. The finding that the addition of talin VBS1, which activates vinculin, results in vinculin binding to polymerizing actin filaments is also neither new and thus not surprising. The quantification of the speed of active vinculin binding to the cytoskeleton is a new detail in the field and perhaps of importance from a mechanistic point of view of nascent adhesion formation.

Likewise, the observations about the robustness of these vinculin-actin interactions is interesting while again the idea that filament elongation and vinculin binding occurring simultaneously have also already been known.

The α-actinin observations with regards to simultaneous binding with vinculin to actin are interesting but perhaps not fully explored and characterized.

How were "rather short filaments" generated?

The analysis of the bundled actin seems problematic where "bridging densities" might be over interpreted as vinculin dimers. Given that the vinculin tail domain binds to actin, I assume the authors are suggesting that the vinculin head domain dimerizes which has not been demonstrated biochemically or structurally. What would cause the vinculin head domain to dimerize? Thompson et al., 2013, suggested that the vinculin tail domain dimerizes but the "bridging densities" do not seem to correlate with such a dimer.

Figure 1A, why 0.6 and 9.3 micromoles of vinculin instead no vinculin at all?

Figure 2A-C is actin alone but 2E analyzes vinculin, where are those actin-vinculin images please?

Figure 2F leaves room for several interpretations.

Figure 3, there seems to be a crucial control missing, actin plus Arp2/3 without vinculin to exclude the possibility that the observed effects are solely due to Arp2/3.

Figure 6, the dimerization is debatable and without any integrin experiments this is also speculative while much of the rest of the model has been published by Waterman's lab although some of her key findings are ignored.

As a side note, the supplementary figures are not labeled and take an effort to track

Figure 1—figure supplement 2, (G) and (H) pull-downs – are all lanes in (G) pellet and are lanes in (H) supernatant and its suggested that only lanes 1-8 are but then what are the other lanes – if they all are pellet (G) and all supernatant (H) then with some actin in the supernatant and vinculin in the pellet, any interpretation would be tricky

I am not certain if Arp2/3 and vinculin binding are sterically hindering each other, surely this depends on the affinity. Arp2/3 is supposed to (biochemical data are missing in the literature) bind the proline-rich region connecting the vinculin head and tail of severed vinculin while the vinculin tail domain binds actin, i.e. two different regions on vinculin harbor the Arp2/3 and actin binding sites.

This study does, however, have some interesting novel aspects. For example, the authors use vinculin that they generated in insect cells which is a plus and new for the field that used to use *E. coli* expressed vinculin. I wonder if there are any remaining residues on this C-terminus after cleavage of the tag and if the C-terminal tag would prevent the closed vinculin conformer.

Videos – what are the arrows pointing to? Without a legend they are difficult to follow.

I am left with the feeling that this manuscript is hard to follow because of too many tangential experiments while a lot has already been published which dilutes the novelty here, that I mentioned above.

Reviewer #3:

The manuscript entitled "Talin-activated vinculin interacts with branched actin networks to initiate bundles" by Paterski et al., describes studies characterizing the underlying mechanism of actin network reorganization and bundling by talin-activated vinculin. Interactions between full-length vinculin and F-actin are monitored, using a simplified in vitro experimental setup to recapitulate actin reorganization in the early events of adhesion assembly. The investigators observe that talin-activated vinculin crosslinks actin filaments into stable and flexible bundles with uniform and mixed polarity using cryo-electron tomography and micro-patterned surface approaches. Further, talin-bound active vinculin is also observed to promote Arp2/3 complex-induced branched actin network organization and dynamics.

Vinculin is a scaffolding protein that localizes to focal adhesions (FA) and adherens junctions where it links the actin cytoskeleton to the adhesive super-structure. It is an abundant and ubiquitously expressed cell adhesion protein that plays a key role in mediating cell adhesion, motility, and cellular response to force. Understanding how talin-mediated activation of full length vinculin promotes actin cytoskeletal rearrangements and how other interacting proteins such as Arp2/3 and actinin modulate these actin networks, will aid in deciphering how vinculin couples the extracellular matrix with the actin cytoskeleton to regulate morphology, motility and mechano-transduction. However, concerns exist with the manuscript, as key controls are missing and some of the data presented does not agree with the existing data/literature. Some of the discrepancies are not acknowledged and discussed.

Detailed comments are listed below:

1) The investigators find that the mean distance between F-actin filaments within the F-actin bundle is 21 +/- 6 nm. This is twice as large as inter-filament distance previously measured by Cryo-electron tomography experiments using isolated Vt domain (10.5 nm) (Janssen, et al., 2006). In the Discussion, authors cite this previous paper without bringing the actual numbers for comparison, and do not elaborate on this difference. However, this difference is crucial for the mechanism of vinculin-mediated F-actin bundling. If the distance between F-actin filaments is increased 2-fold, two Vt domains cannot interact. This would imply that vinculin in the assays should mediate bundling through interaction with linker region or head domain. Indeed, the authors speculate: "Thus, the ~21 nm interspace within vinculin-mediated bundles may represent the length of an open vinculin protein suggesting that vinculin's head domain may play a role in the dimerization process to produce an activated full-length vinculin dimers." However, this is counter to an abundance of evidence in the literature which support actin binding and dimerization primarily through the tail domain.

The disconnect with published literature raises the following concerns:

i) Does Alexa88-modified full length vinculin alter actin bundling and/or other vinculin interactions? Actin binding but surprisingly not bundling controls are reported for baculovirus expressed full length vinculin but not the fluorescent-tagged protein. A comparison of actin binding and bundling is needed for both non-modified baculovirus expressed and fluorescently-tagged vinculin, in the absence and presence of the talin VBS1 peptide.

ii) Alternatively, could the actin "bundling" observed be due to non-specific association of vinculin head domains (Vh) and linkers connecting Vh to Vt? In cells, the position of Vh is strongly restrained by its binding to cell membrane through talin. The experiments herein are performed on a 2D coverslip, where Vh is free to move and potentially create gel-like structures, bridging F-actin filaments together. As a result, Vh domains may be forced between actin filaments (since they cannot escape in 3D) and may prevent F-actin fibers from coming close enough for Vt domains to interact?

2) The authors observe that vinculin and a-actinin colocalize within F-actin bundles. In support of this observation the authors argue that a-actinin and vinculin supposedly have compatible inter-filaments spacing (~20 nm). This conclusion contradicts published literature (P. Kanchanawong, et al., Nanoscale architecture of integrin-based cell adhesions. Nature 468, 580-4 (2010); L. B. Case, et al., Molecular mechanism of vinculin activation and nanoscale spatial organization in focal adhesions. Nat. Cell Biol. 17, 880-892 (2015)), in which a-actinin and vinculin mostly occupy distinct layers, consisting of a membrane-associated integrin signaling layer, a force transduction layer containing talin and vinculin, and an actin-regulatory layer containing a-actinin as well as zyxin and VASP.

3) Known vinculin activation, actin binding and bundling mechanisms should be discussed, to put the data into context. For example, upon vinculin activation (which is still not completely understood), actin binding to Vt promotes a conformational change in the tail domain that facilitates actin dimerization and bundling. Moreover, actinin engages the head domain and can promote Vt activation in a manner similar to talin (Bois et al., JBC, 2006), which may be why these activating ligands localize to different “layers” within the focal adhesion. Arp2/3 engages the proline rich domain and may not compete with direct actin binding interactions through the tail domain. As mentioned above, in cells, vinculin has been shown to localize to different pools, which are “lost” in the simplified in vitro system used here. Thus, the investigators need to extrapolate their findings to a physiologically relevant system whereby vinculin localization occurs with distinct ligands.

4) The authors state that vinculin-mediated F-actin bundles have mixed polarity. The quantity plotted at Figure 2E is (MPs-UPs)/(MPs+UPs), where MPs describes degree of mixed polarity of neighboring fibers, and UPs describe degree of uniform polarity. However, the investigators do not adequately provide a definition of MPs and UPs. Figure 2F depicts crosslinking bridge between the actin filaments. Can the authors elaborate on the orientation of vinculin at these crossing junctions? If bundling is mediated by the tail domain, it is unclear how Vt can dimerize to form a mixed bundle. Moreover, while the mixed polarity observation is intriguing, it is unclear how this fits with published findings that vinculin forms directionally asymmetric catch bonds with F-actin (Huang et al, 2017). Such "mixed polarity bundles" should be very unstable in vivo and shear apart under force.

5) It is apparent from Figure 2D that the actin bundles, except bundle-5, show mixed polarities. However, Figure 2E shows bundles 4,5, and 7 that exhibit uniform polarity. It would be helpful to clarify the apparent discrepancy between Figure 2D and 2E. Moreover, bundles 5 and 7 in Figure 2D depict long and uniform filaments of actin in sharp contrast to bundles 4 and 16. Some discussion is needed to explain these differences.

6) The authors observe that vinculin doesn't enhance F-actin polymerization rate. This observation is also in contradiction with previously published results (Wen et al, 2009; Jannie et al., 2015) which showed that vinculin promotes F-actin polymerization.

7) The vinculin-actin interaction model presented in the Figure 6 seems a bit misleading as it gives the impression that vinculin forms a dimer at the talin interface through its head domain. Actin binding to vinculin has been shown to be mediated primarily through the tail domain. Some discussion is needed here.

8) All proteins are conjugated with covalent fluorophores with varying degree of labeling. And since various parameters noted in the paper are calculated based on the fluorescence quantification coming from these fluorophores, it would be helpful if the authors tabulate the protein with respective fluorophore and degree of labelling and their effective percentages used in fluorescence ratio calculation for each experiment.

Reviewer #4:

Boujemaa-Paterski and colleagues use a minimal in vitro system to mimic situations in cells where initial adhesion complexes become linked to the branched actin network in protruding lamellipodia of migrating cells.

Major findings are that only vinculin activated by talin is able to bind and bundle actin filaments of mixed population; another focal adhesion protein, a-actinin1, is attracted at later stages. When immobilised on micropatterns, activated vinculin traps and organises (bundles) Arp2/3 induced branched actin filaments. At first, many of these individual aspects may not seem entirely novel but the comprehensive way of combining cutting edge technology reveals in a very clear manner how individual components, following specific activation steps, act together to tether initial adhesion complexes to the actin network.

Points to consider:

1) Authors speculate that vinculin dimerization involving its head domain may be important for the bundling function but they do not show this. A potential experiment to clarify could be to use the actin binding vinculin tail only for their experiments.

2) Vinculin itself binds to the Arp2/3 complex. Does this interaction have a role in trapping the Arp2/3 induced branched network?

3) A limitation of the study may be it has not included talin itself, which binds and organises actin prior to vinculin. The presence of talin could change protein binding kinetics and aspects of actin organisation. New experiments may be beyond the scope of this study but this matter requires discussion in the relevant section (see comment point 5).

4) There is no statistics to the histogram in Figure 3A.

5) The Discussion remains superficial in relevant aspects:

a) In a somewhat similar study, Ciobanasu et al., 2014, have shown how the talin-vinculin complex leads to actin anchoring and a-actinin contributes to the crosslinking (bundling) of the initial network. Questions arising are, whether for example kinetics of a-actinin (and vinculin) binding to filaments or the distance between filaments etc. would change in presence of full-length talin?

b) A recent study (Atherton et al. JCB) shows that the release of an autoinhibition motif in talin unmasks VBSs that recruit and activate vinculin. The finding seems in line with the present study showing that an isolated VBS can activate vinculin to bind to actin and should be discussed.

c) There is little discussion about the potential role of Arp2/3 binding to vinculin (DeMali et al., 2002 and other manuscripts). Could the Arp2/3 binding site in vinculin contribute to the connection to the actin meshwork?

[Editors’ note: further revisions were suggested prior to acceptance, as described below.]

Thank you for submitting your article "Talin-activated vinculin interacts with branched actin networks to initiate bundles" for consideration by *eLife*. Your article has been reviewed by four peer reviewers, one of whom is a member of our Board of Reviewing Editors, and the evaluation has been overseen by Suzanne Pfeffer as the Senior Editor. The following individual involved in review of your submission has agreed to reveal their identity: Christoph Ballestrem (Reviewer #4).

The reviewers have discussed the reviews with one another and the Reviewing Editor has drafted this decision to help you prepare a revised submission.

The reviewers have agreed that the revised paper is a substantial improvement over the original submission, and is close to being ready for acceptance in *eLife*. However, in a somewhat unusual move, the reviewers thought that the final paper should contain less data than what is currently shown (rather than more), due to the major concerns about Figure 3—figure supplement 2. Since it was felt that the biochemistry shown here is not essential to the paper, the easiest path is simply to have you remove it rather than asking for a major revision where this work would be redone.

One reviewer stated:

Panels F and H cannot be unambiguously interpreted:

Their pull-down shows that their vinculin aggregates and thus pellets so one cannot say anything about vinculin binding to F-actin if the readout is its pelleting, as it pellets in the absence of F-actin also. More complications are caused by F-actin not completely being polymerized as they also have actin in the supernatant. To me, these data are thus uninterpretable. However, they get the assay to work in panel G where vinculin is not aggregating, and actin does not seem to be unpolymerized.

Since the other reviewers are positive about this manuscript and since these panels are not even mentioned in the manuscript, one way to reconcile the three reviewers would be to take those panels out.

Also please correct this issue and clarify the text as needed:

With regard to Figure 3—figure supplement 2, it is concerning that full length vinculin pellets in the presence of VBS1 but in the absence of actin (vinculin oligomerization, stability issues?), and that actin is present in the soluble fraction upon polymerization. This raises additional concerns on my end regarding use of purified vinculin for their assays. Also, in Materials and methods, it is stated that the pelleting speed is 380,000, but in the figure legend, the pelleting speed is stated as 80,000.

---

## [Author Response]

[Editors’ note: The authors appealed the original decision. What follows is the authors’ response to the first round of review.]

Reviewer #1:This is a rather complicated study that aims to provide new insights into how talin and vinculin interact with actin to form bundles in cells. The work done is purely in vitro, which is necessary to simplify and control the system. While some aspects appear novel, I will need to defer to other reviewers in terms of the biological significance of the results. I am not bothered at all about the seeming conflicts with Janssen et al., 2006, since Kim et al., 2016, came to very different conclusions. The work on actin polarity appears to have been done well and is convincing.

We thank the reviewer for his insightful comments and for his appreciation of the work.

We reorganized the manuscript in order to focus the manuscript around the central claim, namely the ability of activated vinculin to reorganize lamellipodial-like branched actin networks.

We hope that displaying vinculin’s ability to affect such networks from the mesoscale, to the macroscale, down to the filament scale will make our study less complicated and easier to follow.

Our results obtained with single-filament TIRF microscopy as well as the structural analysis of our in vitro vinculin-F-actin bundles using cryo-ET and image processing are now used as supplementary information to the reorganization of Arp2/3 complex-branched followed by TIRF microscopy.

Reviewer #2:The finding that vinculin does not decorate actin without vinculin activation is not novel and thus not surprising. The finding that the addition of talin VBS1, which activates vinculin, results in vinculin binding to polymerizing actin filaments is also neither new and thus not surprising. The quantification of the speed of active vinculin binding to the cytoskeleton is a new detail in the field and perhaps of importance from a mechanistic point of view of nascent adhesion formation.

We agree with the reviewer’s concerns raised concerning the novelty of the ability of talin-VBS1 to relieve vinculin autoinhibition (Papagrigoriou et al., 2004). Our study focuses on the ability of vinculin to bind and bundle branched actin network. For that purpose, we used the published knowledge and activated vinculin in order to study its intrinsic properties with regard to actin binding and bundling.

To the best of our knowledge the quantifications (using full length protein) are novel and may explain the efficacity at which talin-vinculin complexes can engage and reorganize lamellipodial branched-actin networks at the nascent adhesions.

Likewise, the observations about the robustness of these vinculin-actin interactions is interesting while again the idea that filament elongation and vinculin binding occurring simultaneously have also already been known.

We would like to emphasize here that there was a discrepancy around the question of how vinculin affect actin polymerization.

Using spectroscopic measurements, the isolated vinculin tail domain (Vt, only 18% of the full length protein) was first shown to activate actin polymerization (Wen et al, 2009). This was contradicted by LeClainche et al., who proposed that Vt specifically blocks actin filaments barbed end elongation via its C-terminal amino-acids. Namely, it leads actin filaments to elongate only from their pointed ends. In addition, the study also showed that full length vinculin expressed in bacteria (and activated by VBS1) fails to block the barbed end of actin filament (Le Clainche et al., 2010).

We acknowledged the discrepancy in our revised manuscript. Yet, the dynamics of vinculin-F-actin bundles formation, the dynamics and the orientation of filaments within the bundles, the flexibility of bundles were not yet reported. Therefore, these are novel and important basic characteristics of vinculin.

The α-actinin observations with regards to simultaneous binding with vinculin to actin are interesting but perhaps not fully explored and characterized.

We agree with the reviewer’s comment. We fell that an in-depth comparison of the effects of vinculin and α-actinin is beyond the scope of our study, therefore, the α-actinin observations were removed.

How were "rather short filaments" generated?

To preform short actin filaments, we first polymerized actin monomers in the presence of the capping protein α1β2, CP, with a 1:200 molar ratio CP:actin. Given that CP binds actin filament barbed ends fast and stably (Kd ≈ 0.1 nM, Schafer et al., 1996), we expected CP-capped actin filaments to measure 0.7 µm in average.

This information is now provided in the revised manuscript, Figure 3—figure supplement 4 legend and in the revised Material and methods section.

The analysis of the bundled actin seems problematic where "bridging densities" might be over interpreted as vinculin dimers. Given that the vinculin tail domain binds to actin, I assume the authors are suggesting that the vinculin head domain dimerizes which has not been demonstrated biochemically or structurally. What would cause the vinculin head domain to dimerize? Thompson et al., 2013 suggested that the vinculin tail domain dimerizes but the "bridging densities" do not seem to correlate with such a dimer.

We agree with the reviewer that only structural determination would allow an unambiguous statement about the vinculin bridges, seen in our tomograms. However, since vinculin does not seem to form homogeneous coating of the actin, we could not identify the structure of vinculin within the averaged structures. Nevertheless, we preformed additional experiments to identify vinculin structures and locations within tomograms of vinculin-induced actin bundles.

First, we increased the density of bridges by activating vinculin with GFP-talin-VBS1 construct. Second, we used ~2.4 nm gold-nanoparticles (AuNPs) that were synthetized, coupled to talin-VBS1 and used to activated vinculin and induced bundles. These experiments were added as a Figure 3—figure supplement figure 3 and the technical details can be found in the Material and methods section.

As expected, the densities observed between the filaments (“bridges”) are irregularly distributed along the actin as well as the gold-labelled VBS1.

As a comparison, we reconstructed actin bundles using the Vt (Figure 3—figure supplement 3A and Material and methods section).

Figure 1A, why 0.6 and 9.3 micromoles of vinculin instead no vinculin at all?

Control experiments are now provided in Figure 3—figure supplement 1.

Figure 2A-C is actin alone but 2E analyzes vinculin, where are those actin-vinculin images please?

The figure indeed showed bundles which were assembled in the presence of talin-VBS1-activated vinculin.

In the new experiments, the occurrence of these densities was increased. This information is provided in Figure 3—figure supplement 3.

Figure 2F leaves room for several interpretations.

We agree that in a context where vinculin is not attached to a surface, the inter-filament spacing we measured may reflect interactions within vinculin dimer, as previously depicted by Molony and Burridge, 1985, or of a different nature, and that may not be seen in vivo. However, since this is not the major aim in this study, we avoid this analysis.

In this analysis, we rather focused on determining the polarity of filaments within vinculin-F-actin bundles, which to our understanding is a key intrinsic property of vinculin that explains its ability to bundle the intermeshed, branched lamellipodial actin network. However, it is plausible that previous experiments using Vt (187 aa) may not represent the precise functional mechanism of full-length vinculin (1066 aa).

Figure 3, there seems to be a crucial control missing, actin plus Arp2/3 without vinculin to exclude the possibility that the observed effects are solely due to Arp2/3.

We have added the requested controls and quantifications. No actin bundling in the absence of activated-vinculin was detected. The information is provided in Figure 3.

Figure 6, the dimerization is debatable and without any integrin experiments this is also speculative while much of the rest of the model has been published by Waterman's lab although some of her key findings are ignored.

We would like to mention that the work of Waterman’s lab was of course cited also in the previous version. However, our revised manuscript stressed the fact that Thievessen et al., 2013 have shown that vinculin tail interacts with lamellipodial actin and affects nascent adhesions turnover. We also highlighted that due to actin-generated tension vinculin adopts an extended conformation (Case et al., 2015). This can be found in the revised manuscript. Here, we show direct interactions between the proteins, while inside cells it is much more challenging to identify such direct interactions.

Moreover, Thievessen et al. paper stated that “…As the resolution of our TFM was not sufficient to analyze nascent FA, we restricted our analysis to mature FA…”. In addition, Xu et al. 2012 Nat. Methods, reported that investigation of the lamellipodial actin was hindered due to actin density and diffraction-limited scale of nascent adhesions.

While cellular study is of a major importance, biochemical and dynamics properties in vitro are fundamental to understand cellular behavior in details. In our study, vinculin interactions with actin are investigated in in vitro systems that may be relevant for the early-stage of nascent adhesion formation. As discussed in the revised manuscript, several lines of evidence (Bachir et al., 2014 ; Atherton et al., 2020; Dedden et al., 2020 ; Kelley et al., 2020) show that talin-vinculin precomplex can form before associating with integrins. On the basis of our reconstitution experiments, we propose that these precomplexes may stably interact with branched lamellipodial actin, and bundles can be initiated in a force independent manner. The formation of bundles from lamellipodial actin may stabilize nascent adhesions, which is necessary for their maturation into focal adhesions. Additionally, our in vitro reconstitution experiments unambiguously demonstrate that the reorganization of the lamellipodial network induced by vinculin slow down its progression (Figure1, in agreement with in vivo reports by Choi et al., 2008 and Thievessen et al., 2013) which may serve as a key step for the formation of a stable connection between actin – talin-vinculin, at nascent adhesions.

As a side note, the supplementary figures are not labeled and take an effort to trackFigure 1—figure supplement 2, (G) and (H) pull-downs – are all lanes in (G) pellet and are lanes in (H) supernatant and its suggested that only lanes 1-8 are but then what are the other lanes – if they all are pellet (G) and all supernatant (H) then with some actin in the supernatant and vinculin in the pellet, any interpretation would be tricky.

We apologize for the inconvenience. All figures are labelled. The figure legend has been rephrased and the revised figure is now Figure 3—figure supplement 2.

I am not certain if Arp2/3 and vinculin binding are sterically hindering each other, surely this depends on the affinity. Arp2/3 is supposed to (biochemical data are missing in the literature) bind the proline-rich region connecting the vinculin head and tail of severed vinculin while the vinculin tail domain binds actin, i.e. two different regions on vinculin harbor the Arp2/3 and actin binding sites.

Our quantifications of the binding of Arp2/3 and vinculin alone or together indicated that the differences fall within the standard deviation of measurements. Thus, we do not see significant hindering of one in the presence of the other.

We also did not record any activation of Arp2/3 complex by vinculin. This suggest that vinculin may interact with a cellular subpopulation of Arp2/3 (Chorev et al., 2014) or an additional factor is needed for the two to form a complex.

The information is provided in revised Figure 3.

This study does, however, have some interesting novel aspects. For example, the authors use vinculin that they generated in insect cells which is a plus and new for the field that used to use *E. coli* expressed vinculin. I wonder if there are any remaining residues on this C-terminus after cleavage of the tag and if the C-terminal tag would prevent the closed vinculin conformer.

We thank the reviewer for the supportive comment.

Indeed, the vinculin was expressed and purified from insect cells is in a close conformation, as we previously determined its structure by X-ray crystallography (Chorev et al., 2018).

Videos – what are the arrows pointing to? Without a legend they are difficult to follow.

We apologize for missing the legends for these videos. Legend are now provided.

I am left with the feeling that this manuscript is hard to follow because of too many tangential experiments while a lot has already been published which dilutes the novelty here, that I mentioned above.

We agree with the reviewer and focused the manuscript in order to make its message clearer.

Reviewer #3:The manuscript entitled "Talin-activated vinculin interacts with branched actin networks to initiate bundles" by Paterski et al., describes studies characterizing the underlying mechanism of actin network reorganization and bundling by talin-activated vinculin. Interactions between full-length vinculin and F-actin are monitored, using a simplified in vitro experimental setup to recapitulate actin reorganization in the early events of adhesion assembly. The investigators observe that talin-activated vinculin crosslinks actin filaments into stable and flexible bundles with uniform and mixed polarity using cryo-electron tomography and micro-patterned surface approaches. Further, talin-bound active vinculin is also observed to promote Arp2/3 complex-induced branched actin network organization and dynamics.Vinculin is a scaffolding protein that localizes to focal adhesions (FA) and adherens junctions where it links the actin cytoskeleton to the adhesive super-structure. It is an abundant and ubiquitously expressed cell adhesion protein that plays a key role in mediating cell adhesion, motility, and cellular response to force. Understanding how talin-mediated activation of full length vinculin promotes actin cytoskeletal rearrangements and how other interacting proteins such as Arp2/3 and actinin modulate these actin networks, will aid in deciphering how vinculin couples the extracellular matrix with the actin cytoskeleton to regulate morphology, motility and mechano-transduction. However, concerns exist with the manuscript, as key controls are missing and some of the data presented does not agree with the existing data/literature. Some of the discrepancies are not acknowledged and discussed.Detailed comments are listed below:1) The investigators find that the mean distance between F-actin filaments within the F-actin bundle is 21 +/- 6 nm. This is twice as large as inter-filament distance previously measured by Cryo-electron tomography experiments using isolated Vt domain (10.5 nm) (Janssen et al., 2006). In the Discussion, authors cite this previous paper without bringing the actual numbers for comparison, and do not elaborate on this difference. However, this difference is crucial for the mechanism of vinculin-mediated F-actin bundling. If the distance between F-actin filaments is increased 2-fold, two Vt domains cannot interact. This would imply that vinculin in the assays should mediate bundling through interaction with linker region or head domain. Indeed, the authors speculate: "Thus, the ~21 nm interspace within vinculin-mediated bundles may represent the length of an open vinculin protein suggesting that vinculin's head domain may play a role in the dimerization process to produce an activated full-length vinculin dimers." However, this is counter to an abundance of evidence in the literature which support actin binding and dimerization primarily through the tail domain.

We agree with the reviewer comments, although one should take into consideration that a small part/domain of a protein (Vt,187aa) may not fully reflect the structure and dynamics of a large protein (Vinculin, 1066aa). Therefore, studying the full-length protein, ~117 kDa, is of major importance and relevance in comparison to a ~21 kDa protein segment, the Vt. Moreover, we would like to remind the reviewer here that the ability of vinculin to interact and bundle a branched actin network is the major focus of our work and therefore the actin bundles characterization is moved into the supplementary information.

Nevertheless, we have now reconstituted bundles with Vt and obtained an inter-filament distance of 10 nm, matching the value reported by Janssen et al. We have also developed two additional strategies: increasing the density of vinculin density by activating vinculin with GFP-talin-VBS1 construct, and by using 2-3 nm Gold-nanoparticles that were coupled to talin-VBS1 and used to activate vinculin. This information is added as Figure 3—figure supplement 3 and Material and methods section.

In the context of reconstitution using F-actin and soluble, activated VBS1-vinculin, the inter-filament spacing was 29 nm, and depicted an intrinsic property of activated vinculin. Similarly, the 10 nm inter-filament spacing obtained with the tail is an intrinsic property of the isolated Vt.

We agree that a wealth of evidence extensively characterized in vitro the dimerization of isolated Vt (Johnson and Craig, 1995, Kim et al., 2016, and as reviewed in Thompson et al., 2013). in vivo characterizations measured the conformational change in vinculin (Johnson and Craig, 2000, Case et al., 2015), showing that the open, extended state of vinculin correlates with focal adhesion maturation state. Yet, the dynamics of vinculin’s opening upon its activation by talin as well as its stretching remains unclarified.

Therefore, the inter-filament spacing we reported here might be of interest as it describes an intrinsic property of the full-length protein. Yet, our result does not determine whether the spacing represents a transient intermediate state of vinculin-mediated bundles at the onset of the adhesion formation, nor describe the structure and dynamics of the reconstituted bridges. That is beyond the scope of our study and needs further investigation, although in a recent cryo-ET of analysis (Martins et al., 2020, https://www.biorxiv.org/content/10.1101/2020.03.11.987263v2) we measure a mean distance of 28 nm between filaments at FAs in intact cells.

We clarified our statement as mentioned in the revised manuscript.

We have also moved this information in the supplementary information of Figure 3, which describes the branched network reorganization at the single-filament scale. The revised figure label is Figure 3—figure supplement 4.

The disconnect with published literature raises the following concerns:i) Does Alexa88-modified full length vinculin alter actin bundling and/or other vinculin interactions? Actin binding but surprisingly not bundling controls are reported for baculovirus expressed full length vinculin but not the fluorescent-tagged protein. A comparison of actin binding and bundling is needed for both non-modified baculovirus expressed and fluorescently-tagged vinculin, in the absence and presence of the talin VBS1 peptide.

We have added a time-lapse sequence showing that unlabeled vinculin expressed insect was unable to bundle actin filaments in the absence of the talin VBS1 domain, while addition of talin-VBS1 activated its bundling activity.

As we reported bundling activity also with unlabeled proteins, we concluded that the bundling observed with labeled vinculin was not due to any perturbation resulting from chemical labeling by Alexa-488.

This information is provided in Figure 3—figure supplement 1.

The reviewer would agree that expressing vinculin (a eukaryotic protein) in eukaryotic expression system is much more reliable than the expression of mammalian protein in prokaryotes, as previously done.

ii) Alternatively, could the actin "bundling" observed be due to non-specific association of vinculin head domains (Vh) and linkers connecting Vh to Vt? In cells, the position of Vh is strongly restrained by its binding to cell membrane through talin. The experiments herein are performed on a 2D coverslip, where Vh is free to move and potentially create gel-like structures, bridging F-actin filaments together. As a result, Vh domains may be forced between actin filaments (since they cannot escape in 3D) and may prevent F-actin fibers from coming close enough for Vt domains to interact?

The experiments in solution are designed to characterize the basic properties of full-length vinculin. In physiological conditions, vinculin binds activated talin, which is attached to integrin (Case and Waterman, 2015) or membrane-associated PIP2 (Kelley et al., 2020), and therefore it is only activated close to the membrane. This property is demonstrated in Figure 4, which shows that bundling of actin occurs when vinculin is indirectly attached to the surface (via talin VBS1). At FAs, the close proximity of vinculin binding domain in talin may force the head domains to be very close to each other, as well as the localization of many talin proteins in close proximity. Therefore, the spatial proximity of the Vh domain at adhesion sites is dictated, while the bulk experiments may force such a proximity in another manner. In any case, in the revised manuscript we removed the claim that Vh-Vh form a dimer, despite the electron microscopy images provided by Molony and Burridge, 1985

We agree with the reviewer that the interactions and affinity between Vh-Vh domains is of prime interest but it is beyond the scope of our study.

2) The authors observe that vinculin and a-actinin colocalize within F-actin bundles. In support of this observation the authors argue that a-actinin and vinculin supposedly have compatible inter-filaments spacing (~20 nm). This conclusion contradicts published literature (Kanchanawong et al., 2010; Case, et al., 2015), in which a-actinin and vinculin mostly occupy distinct layers, consisting of a membrane-associated integrin signaling layer, a force transduction layer containing talin and vinculin, and an actin-regulatory layer containing a-actinin as well as zyxin and VASP.

Due to the comments from reviewer #1 we removed the a-actinin experiments. However, our data does not contradict Dr. Waterman’s findings. In fact, several studies have detected α-actinin at the onset of adhesion assembly, at the stage of nascent ones. Choi et al., 2008 and Bachir et al., 2014, show that α-actinin, concomitantly to vinculin and talin, transiently associates to nascent adhesions during their stabilization and growth phases. These findings do not exclude nor contradict a later recruitment of α-actinin to the acto-myosin associated fibers.

3) Known vinculin activation, actin binding and bundling mechanisms should be discussed, to put the data into context. For example, upon vinculin activation (which is still not completely understood), actin binding to Vt promotes a conformational change in the tail domain that facilitates actin dimerization and bundling. Moreover, actinin engages the head domain and can promote Vt activation in a manner similar to talin (Bois et al., 2006), which may be why these activating ligands localize to different “layers” within the focal adhesion. Arp2/3 engages the proline rich domain and may not compete with direct actin binding interactions through the tail domain. As mentioned above, in cells, vinculin has been shown to localize to different pools, which are “lost” in the simplified in vitro system used here. Thus, the investigators need to extrapolate their findings to a physiologically relevant system whereby vinculin localization occurs with distinct ligands.

We thank the reviewer for this remark. Following these suggestions, we extrapolated our findings to physiologically relevant scenarios and extended our discussion, as mentioned in the revised manuscript, Discussion section.

4) The authors state that vinculin-mediated F-actin bundles have mixed polarity. The quantity plotted at Figure 2E is (MPs-UPs)/(MPs+UPs), where MPs describes degree of mixed polarity of neighboring fibers, and UPs describe degree of uniform polarity. However, the investigators do not adequately provide a definition of MPs and UPs.

We apologize for the unclear term. We have properly defined it in the revised manuscript. Uniform-polarity score (UPs) and mixed-polarity score (MPs) are now provided in the Materials and methods section and in the figure legend Figure 3—figure supplement 4.

Figure 2F depicts crosslinking bridge between the actin filaments. Can the authors elaborate on the orientation of vinculin at these crossing junctions?

In order to reveal the orientation of vinculin along actin, a uniform interaction (binding) along actin filaments is needed. However, it seems that full-length vinculin cannot saturate all sites on actin presumably because of the size of the protein, therefore our trials to determine the structure were unsuccessful.

If bundling is mediated by the tail domain, it is unclear how Vt can dimerize to form a mixed bundle. Moreover, while the mixed polarity observation is intriguing, it is unclear how this fits with published findings that vinculin forms directionally asymmetric catch bonds with F-actin (Huang et al, 2017). Such "mixed polarity bundles" should be very unstable in vivo and shear apart under force.

We do agree with the reviewer’s comment and with this respect we have expanded our Discussion, as mentioned in our revised manuscript.

We speculate that vinculin’s early interactions with the branched network (which comprises branches randomly oriented to each other) are fast and enable the protein to assemble random-polarity bundles. This may be a key mechanism that potentiate the connection to mobile lamellipodial network, which in turn applies a selection pressure reinforcing only uniform polarity bundles and further stabilization and maturation of the nascent adhesions.

Moreover, we have found that in intact cells focal adhesion-associated actin has non-uniform polarity (Martins et al., 2020)

5) It is apparent from Figure 2D that the actin bundles, except bundle-5, show mixed polarities. However, Figure 2E shows bundles 4,5, and 7 that exhibit uniform polarity. It would be helpful to clarify the apparent discrepancy between Figure 2D and 2E. Moreover, bundles 5 and 7 in Figure 2D depict long and uniform filaments of actin in sharp contrast to bundles 4 and 16. Some discussion is needed to explain these differences.

We apologize for being unclear. We now better defined MPs and UPs scores in our revised manuscript. A clearer information is provided in the Materials and methods section and in the figure legend Figure 3—figure supplement 4.

The histogram initially Figure 2E, in the revised manuscript Figure 3—figure supplement 4E, shows a polarity score of filament’s neighborhood. While in most bundles non-uniform polarity was found, the score between the different polarities was variable, covering all possibilities.

In brief, for every actin segment we extracted the polarities of the (up to) three nearest segments of neighboring filaments, within a 3D distance of 40 nm, defining its neighborhood. Based on that we determined for each segment the degree of uniform polarity (UP) and mixed polarity (MP) of its neighborhood (Figure 3—figure supplement 5C). This describes the microenvironment of an actin segment. Next, we defined a score (MPs-UPs)/(MPs+UPs) to characterize the overall polarity of each bundle.

The histogram shows that bundles 4 and 5 have a polarity ratio ≈ -0.2 meaning that ~20% of the segments display an opposite polarity with their closest neighbors (alternance blue/red filaments) and ~80% of them have a uniform polarity (packing blue/blue or red/red filaments), as depicted the directionality maps of these bundles in panel D. Thus, both polarities were detected in most bundles.

6) The authors observe that vinculin doesn't enhance F-actin polymerization rate. This observation is also in contradiction with previously published results (Wen et al., 2009); Jannie et al., 2015) which showed that vinculin promotes F-actin polymerization.

We thank the reviewer for this comment. The two papers mentioned used only the tail (187aa out of 1066aa). Therefore, it may not be not surprising that using a full-length protein, expressed in eukaryotic system represent a more realistic system.

Moreover, Leclainche et al., 2010 already contradicted the above-mentioned literature.

We acknowledged the discrepancy in our revised manuscript.

7) The vinculin-actin interaction model presented in the Figure 6 seems a bit misleading as it gives the impression that vinculin forms a dimer at the talin interface through its head domain. Actin binding to vinculin has been shown to be mediated primarily through the tail domain. Some discussion is needed here.

We thank the reviewer for this comment. We modified the model accordingly and expanded our Discussion.

8) All proteins are conjugated with covalent fluorophores with varying degree of labeling. And since various parameters noted in the paper are calculated based on the fluorescence quantification coming from these fluorophores, it would be helpful if the authors tabulate the protein with respective fluorophore and degree of labelling and their effective percentages used in fluorescence ratio calculation for each experiment.

We thank the reviewer for this valuable comment. Accordingly, we calibrated the fluorescence intensities and we are now able to provide apparent equilibrium dissociation constant that better characterize the affinity of vinculin to the different actin organizations. This information is provided in revised Figures 1 and 2, and the Materials and methods section.

Moreover, the cryo-ET measurements were conducted with unlabeled proteins.

Reviewer #4:Boujemaa-Paterski and colleagues use a minimal in vitro system to mimic situations in cells where initial adhesion complexes become linked to the branched actin network in protruding lamellipodia of migrating cells.Major findings are that only vinculin activated by talin is able to bind and bundle actin filaments of mixed population; another focal adhesion protein, a-actinin1, is attracted at later stages. When immobilised on micropatterns, activated vinculin traps and organises (bundles) Arp2/3 induced branched actin filaments. At first, many of these individual aspects may not seem entirely novel but the comprehensive way of combining cutting edge technology reveals in a very clear manner how individual components, following specific activation steps, act together to tether initial adhesion complexes to the actin network.Points to consider:1) Authors speculate that vinculin dimerization involving its head domain may be important for the bundling function but they do not show this. A potential experiment to clarify could be to use the actin binding vinculin tail only for their experiments.

We thank the reviewer with this comment.

Although we tuned down this part of the manuscript and discussed in greater details the bundling of branched actin network, we cloned the tail (879-1066) from our human vinculin construct. Reconstitution of actin bundles and structural analysis by cryo-ET showed bundles with an inter-filament spacing of 10 ± 1 nm (Figure 3—figure supplement 3 and 4), in agreement with previous reports (Janssen et al., 2006; Kim et al., 2016).

These experiments clearly demonstrated that Vt (187aa) behaves differently than the full-length vinculin (1066aa). There are several vinculin binding domains in talin that would anyhow force Vh domains to be very adjacent to each other. This would result in actin bundling, and is demonstrated in Figure 4.

Additional experiments revealed increased densities, mediated by talin-VBS1-activated full-length vinculin as well as gold labeling, indicating that bundling by vinculin does not form homogeneous structures as the Vt does. As we now discuss in the revised manuscript these observed bridge densities may also reflect an early, transient state of activated vinculin prior to its extension between the integrin and actin layers, and maybe that actin-generated force is needed to fully extent the protein.

2) Vinculin itself binds to the Arp2/3 complex. Does this interaction have a role in trapping the Arp2/3 induced branched network?

Using TIRFM assays we did not record any activation of Arp2/3 complex by vinculin. This information is added in revised Figure 3.

We speculate that either vinculin interacts with a cellular subpopulation of Arp2/3, as Chorev et al. have identified using mass-spectroscopy analysis (Chorev et al., 2014), or additional factors are needed to mediate the interaction. In Chorev et al., the Arp2/3 complex that contained vinculin was missing the p41-ARC protein. The Arp2/3 complex used in our assays comprises 7 subunits and was purified using a WAVE-WA affinity column chromatography.

3) A limitation of the study may be it has not included talin itself, which binds and organises actin prior to vinculin. The presence of talin could change protein binding kinetics and aspects of actin organisation. New experiments may be beyond the scope of this study but this matter requires discussion in the relevant section (see comment point 5).

We agree with the reviewer and we have expanded our discussed accordingly, as mentioned the revised manuscript. In Figure 4, we synthetically mimic a situation in which several VBS domains are in close proximity to each other. This better resembles the full-length talin and its increased concentration at FAs.

4) There is no statistics to the histogram in Figure 3A.

We indeed implemented our quantifications and the differences falls now within the standard deviation of measurements. The information is provided in revised Figure 3.

5) The Discussion remains superficial in relevant aspects:a) In a somewhat similar study, Ciobanasu et al., 2014 have shown how the talin-vinculin complex leads to actin anchoring and a-actinin contributes to the crosslinking (bundling) of the initial network. Questions arising are, whether for example kinetics of a-actinin (and vinculin) binding to filaments or the distance between filaments etc. would change in presence of full-length talin?

We thank the reviewer for his comment and have expanded our Discussion accordingly, as mentioned in the revised manuscript. Interestingly, the distances measured between actin filament in FAs, by means of cryo-ET (Martins et al., 2020 , doi.org/10.1101/2020.03.11.987263), resemble the measure distances in this work.

However, we tuned down our initial work of α-actinin 1, since it was felt insufficient by the other reviewers. We do agree that an in-depth comparison of the dynamics of vinculin and α-actinin 1 is of prime interest, but beyond the scope of the present study.

b) A recent study (Atherton et al. JCB) shows that the release of an autoinhibition motif in talin unmasks VBSs that recruit and activate vinculin. The finding seems in line with the present study showing that an isolated VBS can activate vinculin to bind to actin and should be discussed.

We thank the reviewer for this comment and we have expanded our Discussion accordingly, as mentioned in the revised manuscript.

Additionally, during the revision of our manuscript, Kelley and colleagues (Kelley et al., 2020) used phosphoinositide to assemble in vitro activated talin-vinculin precomplexes in absence of tensile forces. These data are in line with our results and were discussed too.

c) There is little discussion about the potential role of Arp2/3 binding to vinculin (DeMali et al., 2002 and other manuscripts). Could the Arp2/3 binding site in vinculin contribute to the connection to the actin meshwork?

There might be several paths that would allow the interaction between Arp2/3 and vilnculin, however, such interactions were not detected in our experimental setup.

The results shown here and in previous studies by Chorev et al. suggest that lamellipodial actin- associated Arp2/3 and adhesion-associated Arp may be two distinct species.

[Editors’ note: what follows is the authors’ response to the second round of review.]

The reviewers have agreed that the revised paper is a substantial improvement over the original submission, and is close to being ready for acceptance in eLife. However, in a somewhat unusual move, the reviewers thought that the final paper should contain less data than what is currently shown (rather than more), due to the major concerns about Figure 3—figure supplement 2. Since it was felt that the biochemistry shown here is not essential to the paper, the easiest path is simply to have you remove it rather than asking for a major revision where this work would be redone.

We agree with the reviewer assessment that the biochemical experiments aimed at determining the affinity of vinculin to actin filaments are not central to this study.

Additionally, we thank the reviewers for their suggestion and believe that it is fully justified.

Therefore, we have removed panels F to H from Figure 3—figure supplement 2 and have revised the manuscript accordingly.

One reviewer stated:Panels F and H cannot be unambiguously interpreted:Their pull-down shows that their vinculin aggregates and thus pellets so one cannot say anything about vinculin binding to F-actin if the readout is its pelleting, as it pellets in the absence of F-actin also.

We removed these panels and agree with the reviewers that the experiments could have been improved, however, we would like to clarify the following.

When vinculin was incubated with VBS1 in the absence of actin, a fraction of vinculin pelleted using high-speed centrifugation (380,000 g), while vinculin alone cannot be pelleted.

These observations may represent oligomerization of activated vinculin, as previously reported by Molony and Burridge, 1985.

During data analysis we considered these activated vinculin polymers as our background and subtracted it for the purpose of Kd calculations. In any case it was removed from the manuscript.

More complications are caused by F-actin not completely being polymerized as they also have actin in the supernatant. To me, these data are thus uninterpretable. However, they get the assay to work in panel G where vinculin is not aggregating, and actin does not seem to be unpolymerized.

Unpolymerized actin that remained in the supernatants is a fundamental property of actin and will always be observed due to the equilibrium between F-actin and the critical concentration of actin (G-actin). Densitometric measurements showed that the unpolymerized actin ranged between 0.08 to 0.1 µM (as was shown in the first eight lanes where actin was incubated with increasing amount of VBS1-activated vinculin). This is the critical concentration of actin (Pollard, Analytical Biochemistry, 1983) that remains unpolymerized and is in equilibrium with polymerized filaments.

Similarly, sedimentation of filamentous actin in panel G showed unpolymerized actin corresponding to the critical concentration (as was shown in the last two lanes of “Pellets” and “Supernatants” gels). Most of the actin (filaments) went to the pellet except the critical concentration that remained unpolymerized, therefore staying in the supernatant. Thus, the intrinsic property of actin to polymerize above its critical concentration was not affected by the presence of vinculin in the absence or in the presence of VBS1.

Since the other reviewers are positive about this manuscript and since these panels are not even mentioned in the manuscript, one way to reconcile the three reviewers would be to take those panels out.

We thank the reviewer for the suggestion. We have removed all biochemistry data in panels G to H.

Also please correct this issue and clarify the text as needed:With regard to Figure 3—figure supplement 2, it is concerning that full length vinculin pellets in the presence of VBS1 but in the absence of actin (vinculin oligomerization, stability issues?), and that actin is present in the soluble fraction upon polymerization. This raises additional concerns on my end regarding use of purified vinculin for their assays.

We hope the explanation that actin can only polymerize above its critical concentration (an intrinsic property of actin defined by the ratio of its dissociation over association rate constants, k-/k+) and therefore will also be found in the supernatant is convincing. Moreover, VBS1-activated vinculin can be found in the pellet due to some oligomerization states as observed by Molony and Burridge, 1985.

Also, in Materials and methods, it is stated that the pelleting speed is 380,000, but in the figure legend, the pelleting speed is stated as 80,000.

We apologize for the typo mistake and the confusion it brought.

We used high-speed sedimentation assays that were all carried out at 380,000x g.